# Targeting the actin nucleation promoting factor WASp provides a therapeutic approach for hematopoietic malignancies

Guy Biber[1,4], Aviad Ben-Shmuel[1,4], Elad Noy[1], Noah Joseph[1], Abhishek Puthenveetil [1], Neria Reiss[1], Omer Levy[1], Itay Lazar[1], Ariel Feiglin[1], Yanay Ofran[1], Meirav Kedmi[1,2,3], Abraham Avigdor[2,3], Sophia Fried[1] & Mira Barda-Saad [1✉]

Cancer cells depend on actin cytoskeleton rearrangement to carry out hallmark malignant functions including activation, proliferation, migration and invasiveness. Wiskott–Aldrich Syndrome protein (WASp) is an actin nucleation-promoting factor and is a key regulator of actin polymerization in hematopoietic cells. The involvement of WASp in malignancies is incompletely understood. Since WASp is exclusively expressed in hematopoietic cells, we performed *in silico* screening to identify small molecule compounds (SMCs) that bind WASp and promote its degradation. We describe here one such identified molecule; this WASp-targeting SMC inhibits key WASp-dependent actin processes in several types of hematopoietic malignancies in vitro and in vivo without affecting naïve healthy cells. This small molecule demonstrates limited toxicity and immunogenic effects, and thus, might serve as an effective strategy to treat specific hematopoietic malignancies in a safe and precisely targeted manner.

[1] The Mina and Everard Goodman Faculty of Life Sciences, Bar-Ilan University, Ramat-Gan 5290002, Israel. [2] Division of Hematology and Bone Marrow Transplantation, Chaim Sheba Medical Center, Tel Hashomer, Israel. [3] Sackler School of Medicine, Tel-Aviv University, Tel-Aviv, Israel. [4] These authors contributed equally: Guy Biber, Aviad Ben-Shmuel. ✉email: Mira.Barda-Saad@biu.ac.il

Hematopoietic malignancies are a leading cause of cancer incidence and death, with an estimated 690,000 deaths and 1,186,590 new cases worldwide in 2018[1]. Hematopoietic malignancies are a heterogeneous collection of lymphoproliferative or myeloproliferative diseases, which can develop in the bone marrow, blood, lymph nodes, and lymphatic or non-lymphatic organs[2]. These types of malignancies are mainly characterized by uncontrolled cellular proliferation, enhanced motility, and invasiveness into adjacent or distal tissues, allowing cells to spread and invade other tissues in the body. In this study, we focused on the common lymphoproliferative hematopoietic malignancies, leukemia and Non-Hodgkin's lymphoma (NHL)[3,4]. Hematopoietic malignancies accounted for ~10% of all cancer-associated deaths in 2018, with NHL and leukemia as the leading hematological cancers[1].

Current treatments for hematopoietic malignancies are still based mainly on chemotherapy and radiotherapy, and are usually followed by hematopoietic stem cell transplantation in cases of leukemia[5]. These treatments are nonspecific and accompanied by severe side effects including sensitivity to infections, hair loss, chronic fatigue, heart disorders, infertility, and secondary cancers[6]. Immune-based therapies, such as monoclonal antibodies, bispecific antibodies, immune checkpoint inhibitors, immunomodulators, and adoptive cell transfer (ACT), have shown significant clinical benefits, although many patients still fail to respond to these treatments due to primary, adaptive, and acquired resistance[7]. In the case of checkpoint inhibitors, treatment with Nivolumab (anti-PD-1) showed promising clinical results for Hodgkin's lymphoma patients[8] but low complete response rates for NHL patients[9]. Furthermore, modest results were obtained using a combination treatment with anti-CTLA-4 and anti-PD-1 antibodies[9–11]. In addition, treatments with anti-CD19 chimeric antigen receptor-modified T (CAR-T) cells in patients with relapsed acute lymphoblastic leukemia (ALL) or refractory B-cell lymphomas showed limited success[12,13]. Although these new immune-based treatments might help to minimize the severe side effects of chemotherapy and radiotherapy, there are many reports of patients who still develop severe side effects from damage to healthy bystander cells, renal damage, cytokine release syndrome, and graft-versus-host disease (GVHD)[14–16]. Therefore, the development of new drugs to treat hematopoietic malignancies with minimal side effects and higher efficacy is still a critical unmet need.

Hematopoietic malignancies are varied in terms of the origin of the neoplastic cells, and are genetically diverse due to random mutations and chromosomal translocation events resulting from viral infections or genome instability[17]. The formation of neoplasms is a multistep process in which the cells acquire biological capabilities known as hallmarks of cancer, that ultimately transform them into malignant cells[18]. The cytoskeleton, and in particularly the actin-network, are vital in order to carry out these newly acquired enhanced cellular capabilities[19–21]. These hallmarks of cancer enable malignant cells to become autonomously activated[22–24], to independently and uncontrollably proliferate, and to increasingly migrate[19,25], invade, and metastasize to adjacent and distal tissues[26]. These activities are defined as actin-dependent processes[21].

Wiskott–Aldrich Syndrome protein (WASp) belongs to a family of actin nucleation-promoting factors (NPFs) that regulate actin-cytoskeleton network rearrangement by activating the actin-related protein 2/3 (Arp2/3) complex. WASp is exclusively expressed in hematopoietic cells[27], and thus has a unique role in their cellular activities[28–30]. This vital role of WASp can be seen in patients with Wiskott–Aldrich Syndrome (WAS) or X-linked thrombocytopenia (XLT). WAS patients exhibit a broad range of immune deficiencies due to the absence or decrease in WASp

expression, respectively[31]. Although WASp itself has a pivotal role in healthy hematopoietic cell function[32–35], its function in malignant hematopoietic cells is still unclear. Other WASp homologs from the WASp NPF family of proteins, such as Neural Wiskott–Aldrich Syndrome protein (N-WASp), WASp family verprolin homologous protein (WAVE) 1–3, Wiskott–Aldrich syndrome protein and SCAR homologue (WASH), WASp homolog associated with actin, membranes and microtubules (WHAMM), and junction mediating regulatory protein (JMY), are ubiquitously expressed in most healthy cell types, and have a well described role in various types of cancers as key players in mediating molecular pathogenesis[26,36–39].

In our previous studies, we established the molecular mechanism regulating WASp degradation[40]. Following cellular activation, WASp undergoes ubiquitylation on lysine residues 76 and 81, located at the WASp homology-1 (WH1) domain, directing WASp to proteasomal degradation. Furthermore, we demonstrated that WASp-interacting protein (WIP) masks WASp ubiquitylation sites at the WH1 domain, thereby protecting it from degradation[41]. Upon cellular activation, WASp is recruited to the membrane, and is activated by the small G-protein, Cdc42, which binds to its GTPase-binding domain (GBD); concurrently, WASp is phosphorylated on tyrosine 291. This activation releases WASp from its auto-inhibitory state and results in a partial dissociation from the C'-terminus of WIP at the N'-terminal WH1 domain of WASp, while retaining its interaction between the C' terminus of WASp and the N'-terminus of WIP. As a result, WASp is subsequently degraded by the Cbl E3 ubiquitin ligases[40–43]. Due to the constant activation of cancer cells, and the induction of WASp activity, which serves as a key regulator of actin-dependent processes including cellular proliferation, migration, and invasiveness[32–35], WASp may play a critical role in mediating these hallmarks of cancer. Of note, WAS patients exhibit deficiencies in all hematopoietic cellular compartments[33,44]. Thus, although the role of WASp in cancer remains enigmatic, it might serve as an attractive target for treatment of a wide range of hematopoietic malignancies in a selective manner, relative to healthy nonactivated hematopoietic cells.

Here, we performed in silico screening using a machine learning trained predictor to identify SMCs that may interfere with the WIP-WASp interaction at the WH1 domain, thereby exposing WASp to degradation. Out of dozens of high score predicted molecules, the SMC, 8-(2,3-dihydro-1H-inden-2-yl)-1-isobutyl-3-(4-methoxybenzyl)-1,3,8-triazaspiro[4.5]decane-2,4-dione, termed WASp-targeting SMC #13 was selected. We show that this agent specifically binds WASp and not its homologs, induces its degradation, and inhibits key cellular WASp-dependent processes including activation, proliferation, migration, and invasiveness of multiple types of malignant hematopoietic cells, without affecting nonhematopoietic or naïve hematopoietic populations. Furthermore, we demonstrate that this WASp-targeting SMC abolishes the growth of human NHL in an in vivo mouse model.

## Results

**In silico screening for WASp-binding SMCs**. In order to identify possible SMCs that can bind to WASp and disrupt its interaction with WIP, we performed machine learning-enhanced virtual screening of drug-like candidate compounds (Supplementary Fig. 1a).

Since the structural model of the WASp WH1 domain is currently unavailable, a surrogate model was created by using the published NMR model of a recombinant WH1 domain of N-WASp, stabilized via a linker fragment and fusion to a short WIP peptide (PDB ID: 2IFS)[45]. Since N-WASp is the closest homolog

**Table 1 Characterization the physiochemical properties of SMC #13 and drug-likeness potential using the SwissADME online tool.**

| Parameters | |
|---|---|
| Molecular weight | 461.6 g/mol |
| No. of H bond acceptors | 4 |
| No. of H bond donors | 0 |
| Topological polar surface area (TPSA) | 53.09 Å |
| Partition coefficient—Log P | 3.78 |
| Solubility—Log S | −5.44 |
| GI absorption | High |
| BBB permeant | Yes |
| P-gp substrate | No |
| Drug likeness | Yes |
| Bioavailability score | 0.55 |

*GI* Gastrointestinal, *BBB* brain–blood barrier, *P-gp* P-glycoprotein.

of WASp, the N-WASp-WIP model was modified with the use of a virtual mutagenesis tool[46] to obtain a relevant prediction for WASp.

Subsequently, this structural model was used to virtually screen a 2D structure library of ~320,000 drug-like compounds that was obtained from ChemBridge Corporation's "CORE Library compound collection" and NIH MLSMR (Molecular Libraries Small Molecule Repository). Up to 10 different 3D conformations were generated for each drug-like compound using the 'FAST' algorithm (as described in Materials and Methods)[47,48].

Putative binding was evaluated for each conformational model of every drug-like SMC candidate with the surrogate WASp model. First, docking was performed via PatchDock (parameters are specified in the Materials & Methods section)[49,50]. Then the interaction was scored using a machine learning-based classifier. Briefly, the classifier used was trained to distinguish between protein-small molecules pairs that bind each other and pairs that do not, based on docking results. The docking models were each transformed into a numerical representation (i.e., a feature vector) based on the characteristics of the putative protein–ligand model of each pair. By contrasting known protein-SMC pairs (positively labeled samples) with random pairing of SMC and proteins (assumed not to occur, thus labeled as negative examples), the classifier was trained using differentiating docking models that are representative of protein-SMC pairs that occur and those that do not. Using the same docking model to feature vector pipeline in conjunction with the trained classifier, putative binding pairs were similarly scored (with a score of 0 signifying a small molecule predicted to be the least likely to bind to the given protein, and a score of 1 indicating molecules predicted as most likely to bind to the protein). Leading SMC-WASp putative binding pairs were selected. Results were further filtered by discarding any structure with a putative binding site that is not in close (>5 Å) proximity to the WASp degradation site, comprising lysine residues 76 and 81. The leading drug-like molecules were further evaluated using Dreiding-like forcefield structural optimization[51] to redock[52] the candidates and the protein. Virtual screening showed that out of dozens of high score predicted molecules, the SMC, 8-(2,3-dihydro-1H-inden-2-yl)-1-isobutyl-3-(4-methoxybenzyl)-1,3,8-triazaspiro[4.5]decane-2,4-dione (Supplementary Fig. 1b), termed WASp-targeting SMC #13, had the required properties, and was selected for further characterization of specificity and biological relevance. We performed an analysis of drug-likeness potential using the SwissADME online tool[53] and characterized the physiochemical properties of SMC #13 (Table 1). Based on the

predicted characteristics of SMC #13, its pharmacokinetic potential as a feasible drug is summarized in a bioavailability radar chart (Supplementary Fig. 1c). The data for mass spectrometry and NMR of SMC#13 are provided as supplementary Fig. 13a and 13b, respectively. Furthermore, we determined the distribution constant (logD~3.3) of SMC #13 experimentally using HPLC (Supplementary Fig. 1d).

**Specific binding of SMC #13 to WASp.** To validate the in silico binding prediction, we confirmed the specific binding of SMC #13 to the WH1 domain of WASp in whole-cell lysates using Microscale thermophoresis (MST)[54]. We expressed YFP-WASp in HEK 293 T cells, which are nonhematopoietic cells and lack expression of endogenous WASp, but express WIP. Whole-cell lysates (WCL) were prepared containing wt YFP-WASp and tested for MST fluorescence quality validation. A binding curve of SMC #13 to YFP-WASp wt was determined, with an estimated Kd of $30.8 \pm 13.6$ nM (Fig. 1a). Furthermore, we tested the in silico predicted WASp WH1 domain binding site of SMC #13. MST analysis showed elimination of SMC #13 binding to the YFP-WASp ΔWH1 mutant, supporting the WH1 domain as the putative binding site of SMC #13 (Fig. 1b).

Biolayer interferometry (BLI) was used as an orthogonal method to validate the specific binding of SMC #13 to the WH1 domain of WASp. As can be seen in Fig. 1c, clear concentration dependent binding occurs between SMC #13 and YFP-WASp wt, while no concentration dependent binding was observed between SMC #13 and YFP-WASp ΔWH1 (Fig. 1d). In MST experiments, no binding was detected between SMC #13 and WASp homologs such as N-WASp or WAVE2 (Supplementary Fig. 2a and b, respectively). In contrast to N-WASp, WAVE2 lacks a WH1 domain[42,55] and is not expected to bind SMC #13 but was used to control for the possible effect on other WASp family members. Moreover, MST analysis showed neither nonspecific binding of SMC #13 to the YFP tag (Supplementary Fig. 2c) nor irrelevant SMC binding (ROCK inhibitorY-27632) to YFP-WASP wt (Supplementary Fig. 2d). Similarly, BLI analysis demonstrated the lack of off-target binding of SMC #13 to N-WASP (Fig. 1e), WAVE2 (Fig. 1f), and the YFP tag, which was taken as a subtracted reference for the BLI experiments. All of the binding experiments were conducted following quality check using western blot analysis (Supplementary Fig. 3a). Collectively these data show that SMC #13 specifically binds WASp at the WH1 domain, and does not bind other NPF homologs, supporting the selective targeting of this compound to hematopoietic cells.

**WASp-targeting SMC #13 downregulates WASp cellular levels by upregulation of its ubiquitylation.** To verify the biological effect of the selected SMC, T-ALL Jurkat cells were incubated for 24 h with increasing concentrations of SMC #13. Following incubation, WASp cellular concentration levels were compared with those detected in control cells treated with equivalent concentrations of DMSO (vehicle) (Fig. 2a, Supplementary Fig. 11). Incubation of leukemic cells with various concentrations of SMC #13 resulted in a corresponding downregulation of WASp in a dose-dependent manner relative to the control (Fig. 2a, Supplementary Fig. 11). At 40 μM SMC #13, the WASp cellular concentration was markedly reduced after 24 h incubation, and thus we chose this concentration for our further in vitro experiments.

We next examined whether this reduction of WASp levels stems from its enhanced ubiquitylation and consequent degradation. T-ALL cells were incubated with SMC #13, followed by treatment with the proteasome inhibitor, MG132. Cells were lysed, and the whole-cell lysates were immunoprecipitated with anti-WASp antibody and immunoblotted with anti-ubiquitin or

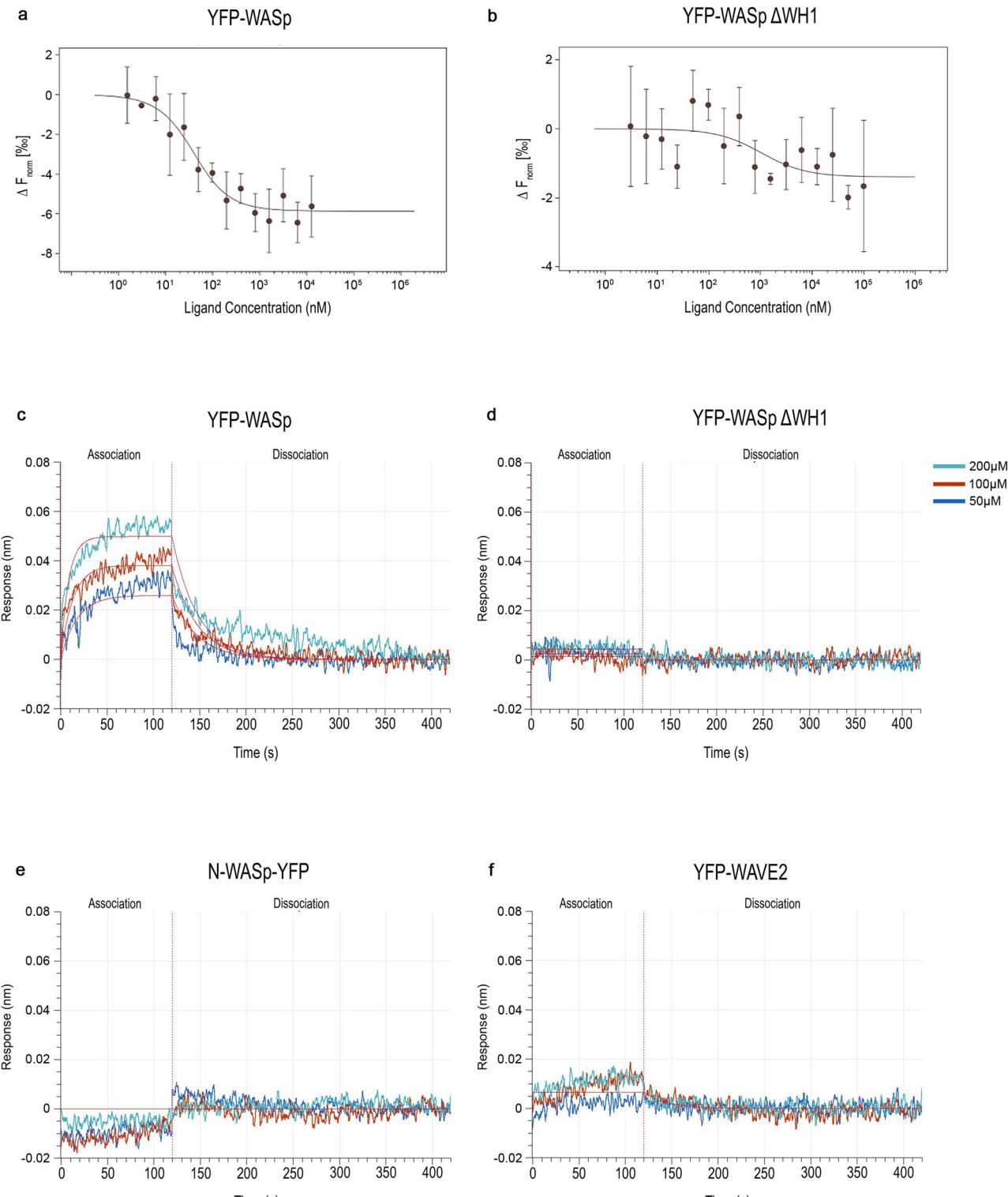

**Fig. 1 Binding of SMC #13 to WASp detected by MST and BLI analyses.** MST: Diluted cell lysates of **a** YFP-WASp wt ($n = 5$), and **b** YFP- WASp ΔWH1 ($n = 5$) from HEK 293 T cells were incubated with serially diluted (1.5 nM–200 μM) SMC #13. The mixed lysates and SMC were then loaded into standard-treated Monolith™ capillaries, and the fluorescence of the samples was measured at 50% blue LED power in the Monolith NT.115 instrument at high MST power. Binding curves were generated by NanoTemper Analysis 2.2.31 software, and the normalized fluorescence is plotted as a function of SMC concentration. The calculated Kd is 30.8 ± 13.6 nM. Data shown as an average ± SD. BLI biolayer interferometry measurements using super streptavidin biosensors with biotinylated GFP antibody showing the binding of SMC #13 to **c** YFP-WASp wt, **d** YFP-WASp ΔWH1, **e** N-WASp- YFP, or **f** YFP-WAVE2. The specific binding of the indicated proteins was analyzed with serially diluted SMC #13 (50–200 μM).

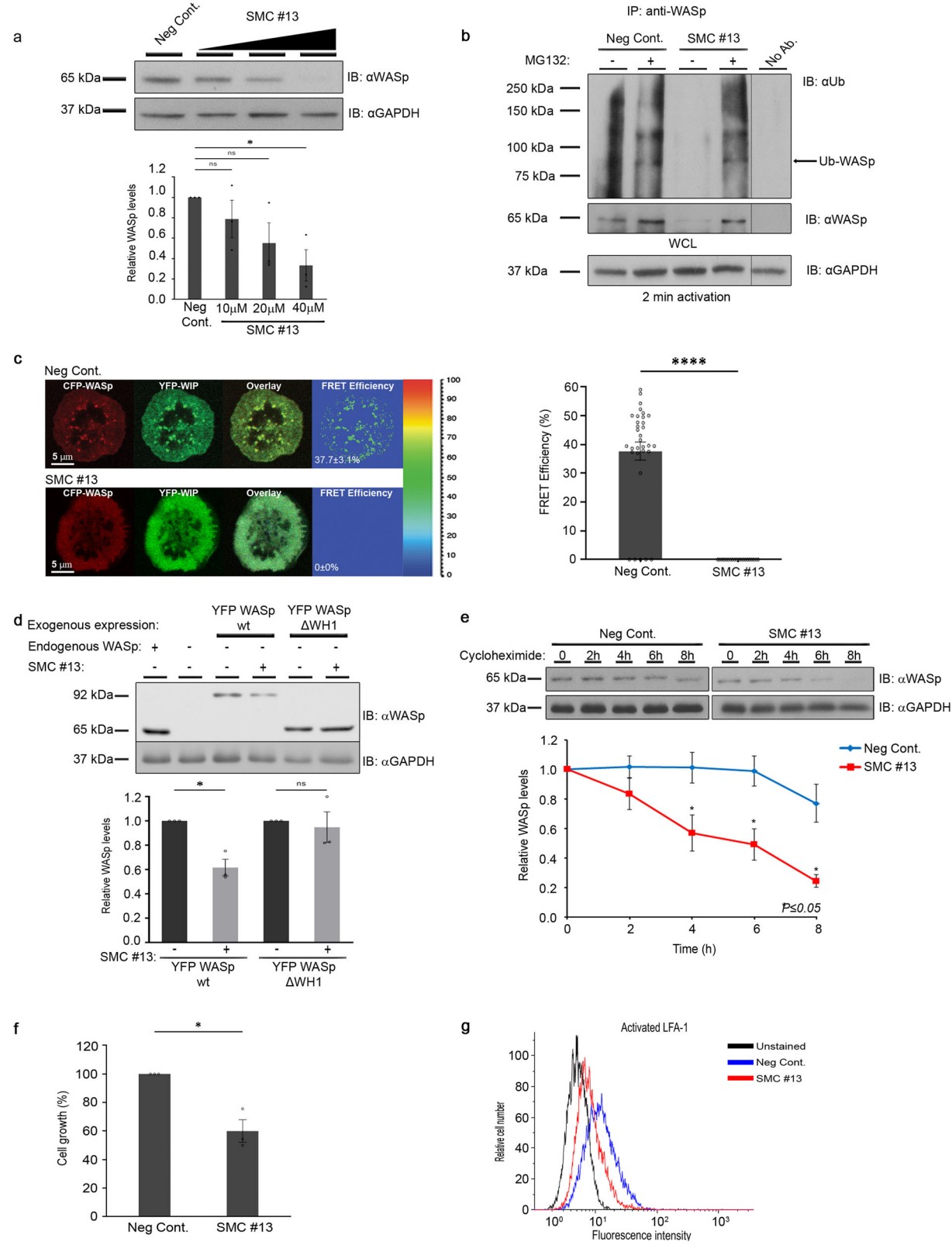

anti-WASp antibodies. As predicted, treatment with SMC #13 almost completely abrogated cellular WASp. However, treatment of the cells with MG132 inhibited WASp degradation and restored ubiquitylated WASp levels (Fig. 2b, Supplementary Fig. 11). These results are consistent with SMC-dependent degradation of WASp through enhanced ubiquitylation and proteasomal-dependent degradation.

WIP binds WASp and protects its degradation sites from ubiquitylation[40]. Thus, we assessed by fluorescence resonance energy transfer (FRET)[41,56–62] whether the enhanced ubiquitylation of WASp following incubation with SMC #13 was the direct consequence of interference with the WIP-WASp interaction. T-ALL Jurkat cells expressing CFP-WASp and YFP-WIP were

**Fig. 2 SMC #13 enhances WASp degradation and downregulates its cellular functions. a** Effect of SMC #13 on WASp levels. T-ALL Jurkat cells were incubated with increasing concentrations, 10 μM (n.s to the control), 20 μM (n.s to the control), 40 μM (p value of 0.04) of SMC #13 or vehicle (as a negative control) for 24 h. Cells were lysed, and WASp cellular concentration (~65 kDa) was analyzed by immunoblot and normalized to GAPDH ~37 kDa as a loading control. Densitometric analysis of the bands was performed with ImageJ. Relative WASp levels were compared to vehicle-treated control cells and are presented in the graph. Values are shown as mean ± standard errors of the mean (SEM) of three independent experiments. The P values were calculated using one-way anova and Dunnetts multiple comparison test. n. s not significant. **b** Ubiquitylation assay. Activated T-ALL Jurkat cells were incubated with 40 μM SMC #13 with (+), or without (−) MG132 proteasome inhibitor. WASp was immunoprecipitated from cell lysates and analyzed by immunoblot for WASp ubiquitylation using anti-Ub Ab. Bands of ubiquitylated WASp appear at and above ~81 kDa. **c** WIP/WASp FRET efficiency. Jurkat cells expressing CFP-WASp and YFP-WIP were incubated with 40 μM SMC #13 or vehicle only, plated on stimulating chambered coverslips coated with anti-CD3 Ab, and stimulated for 2 min before fixation. Activated fixed cells were imaged by Zeiss LSM 510 confocal microscope, and FRET efficiency was measured by donor-sensitized acceptor emission technology. Graph summarizing the mean percentage FRET efficiency. (P value < 0.0001). Values are shown as mean ± SEM (Vehicle, n = 32 cells; SMC #13, n = 20 cells). The P values were calculated using unpaired t-test. n.s not significant. **d** Jurkat cells knocked out for WASp were transfected with either YFP-WASp wt or YFP-WASp ΔWH1. Twenty four hours after transfection, cells were incubated with 40 μM SMC#13 for an additional 24 h. Subsequently, cells were lysed, and lysates were analyzed for YFP-WASp levels by immunoblotting and normalizing to GAPDH loading control. **e** T-ALL Jurkat cells were preincubated with 40 μM SMC #13 or vehicle for 12 h, followed by treatment with 0.1 mg/ml cycloheximide (CHX). Cell samples were collected and lysed at the indicated timepoints. Lysates were immunoblotted with anti-WASp and anti-GAPDH as a loading control (upper panel). Graph summarizing the relative cellular concentration of WASp following incubation with vehicle, or SMC #13 in the presence of CHX (lower panel) (P ≤ 0.05). Values are shown as mean ± standard errors of the mean (SEM). (DMSO, n = 6; SMC#13, n = 3). The P values were calculated using two-tailed t-test. **f** T-ALL Jurkat cells were treated with 40 μM SMC #13 or vehicle for 24 h, and cellular proliferation of the treated cells was determined relative to control by an XTT-based assay (P value = 0.0371). Values are shown as mean ± SEM of three independent experiments. The P values were calculated using two-tailed paired t-test. **g** T-ALL Jurkat cells were incubated with 40 μM SMC #13 or control for 24 h. Cells were then stained with anti-active-LFA-1 integrin Ab (KIM127), followed by staining with Alexa Fluor 488-conjugated goat anti-mouse IgG1 secondary Ab, and analyzed by FACS. Representative flow cytometry plot is presented (P ≤ 0.01).

plated on stimulatory coverslips coated with anti-CD3 antibody, and were then fixed after 2 min of activation[62]. Close association between WIP and WASP is expected to enable high FRET efficiency due excitement of YFP by the CFP emission spectrum. In the control-treated cells, clustering of WIP and WASp occurred with an average FRET efficiency of 37.7 ± 3.1%, while the FRET efficiency between CFP-WASp and YFP-WIP in the SMC #13-treated cells was undetectable (Fig. 2c).

To confirm our results showing that SMC #13 induces downregulation of WASp via interaction with the WH1 domain in hematopoietic cells, we used CRISPR/CAS9 gene editing technology to generate a WASp-deficient Jurkat cell line (Jurkat WASp KO). KO cells were transfected with either YFP-WASp wt or with YFP-WASp ΔWH1. Cells were treated with 40 μM of SMC #13 or DMSO vehicle (control). Incubation with SMC #13 resulted in a reduction of YFP-WASp cellular concentration but not that of YFP-WASp ΔWH1 (Fig. 2d, Supplementary Fig. 11). This result further validates the specificity of SMC #13 for the WASp WH1 domain, which regulates WASp degradation. Moreover, to confirm that WASp degradation is not influenced by induced changes in actin-cytoskeleton polymerization, we incubated Jurkat cells with either SMC #13 or the actin depolymerizing drug, cytochalasin D[63]. Treatment with Cytochalasin D or DMSO (negative control) had no observable effect on WASp stability, whereas a clear reduction in WASp level was observed in cells treated with SMC #13 (Supplementary Fig. 3b). To further confirm the effect of SMC #13 on WASp-post translational modification rather than its transcription, WASp RNA levels were determined in treated and untreated PBMCs from a healthy donor (Supplementary Fig. 3c). No change was observed in the mRNA expression level between the SMC #13 and the vehicle-treated cells (naïve or activated). Thus, these data demonstrate that WASp downregulation is due to enhanced proteosomal degradation rather than alterations in WASp mRNA level or changes in actin cytoskeletal dynamics.

Finally, we further investigated the effect of SMC #13 on WASp protein stability, using a cycloheximide (CHX) chase assay[64]. Short incubation of SMC #13 demonstrated that WASp is significantly degraded after 8 h of incubation (Supplementary Fig. 3d). Therefore, T-ALL cells were pretreated with SMC #13 or vehicle for 12 h. After this pretreatment, cells were also incubated with CHX, and WASp cellular concentration was measured over 8 h (Fig. 2e, Supplementary Fig. 11). In the vehicle-treated cells (neg. cont.) WASp levels were relatively stable, with a noticeable reduction only after 8 h of incubation with CHX. In contrast, in SMC #13-treated cells, WASp was more rapidly degraded and significantly decreased after the first 4 h. WASp levels were reduced following 4, 6, and 8 h by 43.2 ± 12.3% (P ≤ 0.04), 50.9 ± 10.5% (P ≤ 0.02), and by 75.7 ± 4.3% (P ≤ 0.008), respectively. These data demonstrate that SMC #13 downregulates WASp expression on the protein level by a mechanism of enhanced degradation.

**Downregulation of WASp cellular concentration following SMC#13 treatment inhibits the proliferation and integrin activation of cancerous hematopoietic cells.** The proliferation of T cells obtained from WAS patients or WAS-deficient mice is severely decreased[65]. Thus, we tested the effect of WASp degradation by SMC #13 on the proliferation of cancerous cells using the XTT proliferation assay. Treatment with SMC#13 significantly decreased the proliferation of treated T-ALL cells by 40 ± 8% relative to cells treated with vehicle, control (P ≤ 0.05) (Fig. 2f).

Studies using cells from WAS patients or mouse models have shown that WASp is a key regulator of actin-dependent processes, including cell activation, migration, and invasiveness[66–70]. Primary T cells obtained from WAS patients have impaired activation when stimulated with anti-CD3 mAb[24,71,72], and integrin activation[66,73]. We therefore examined the functional consequences of WASp downregulation by SMC #13 on cellular activation, as demonstrated by the affinity maturation of LFA-1 (lymphocyte function-associated antigen 1)[74–77]. Malignant hematopoietic cells are often constitutively activated[23]. This activation is evident by the transition from the low to the intermediate binding affinity conformation of LFA-1 as a result of inside-out signaling[78], which is mainly dependent on F-actin binding to adaptor proteins in the cell membrane, e.g., vinculin or talin[79,80]. The intermediate and high affinity conformations of LFA-1 are recognized by the KIM127 antibody[81,82]. T-ALL cells were treated with SMC #13 (red curve) and analyzed for KIM127 staining using flow cytometry in

comparison to vehicle-treated cells (blue curve). Treatment with SMC #13 dramatically decreased LFA-1 affinity maturation by $62.6 \pm 9.9\%$ ($P \leq 0.01$) (Fig. 2g, Supplementary Fig. 12a).

**WASp-dependent migration is downregulated in hematopoietic cells following SMC #13 treatment**. As mentioned above, WASp is a key factor supporting the directional migration of hematopoietic cells[33,70,83]. To determine the effect of SMC #13 on migration, malignant primary CLL cells and primary NHL cells from marginal B-cell lymphoma patients were obtained from recently diagnosed, untreated patients. Lymphocytes were isolated from the blood samples, and incubated with SMC #13, or with the vehicle as a negative control. WASp cellular concentration was determined by western blot in primary CLL cells (Fig. 3a, Supplementary Fig. 11) and primary NHL cells (Fig. 3b, Supplementary Fig. 11) relative to the vehicle control. SMC #13 caused a reduction of $60 \pm 10.9\%$ ($P \leq 0.032$) and $54 \pm 12\%$ ($P \leq 0.05$) in WASp cellular concentration in primary CLL and NHL cells, respectively. Lymphoma cells are characterized by high migratory capacity, which facilitates their distribution to distant tissues[84]. Since WASp is a key player in cellular motility, we examined whether treatment with WASp-targeting SMC attenuates the migration of primary CLL and NHL cells. The cells were seeded over slides precoated with ICAM-1 and SDF-1α[85,86]. Movement of the cells was automatically tracked and analyzed following treatment with SMC #13[87]. Strikingly, a significant inhibition of cellular migration was observed in SMC #13-treated primary CLL cells, which migrated to a distance of $23.4 \pm 4.4$ μm, compared to control cells, which migrated to a distance of $75.7 \pm 6.7$ μm ($P \leq 0.001$) (Fig. 3c and movies S1-S2). This effect was also evident in primary marginal B-cell lymphoma cells, wherein treatment with SMC #13 significantly decreased cellular motility to $24 \pm 2.5$ μm relative to the control cells, which migrated a distance of $48.1 \pm 2.6$ μm ($P \leq 0.001$) (Fig. 3d and Supplementary movies 3-4). These results were also corroborated using migration analysis of T-ALL expressing WASp wt or WASp KO cells to ensure the on-target effect of SMC #13. As seen in Fig. S3E, WASp KO cells exhibit a heavily reduced cell displacement similar to the well described phenotype of WAS patient cells and similar to SMC#13 treatment[33]. Since WASp is exclusively expressed in hematopoietic cells, we verified that SMC #13 does not affect other WASp family proteins (e.g., N-WASp and WAVEs) in bystander nonhematopoietic cells. Therefore, we used scratch-wound assay to demonstrate that SMC #13 does not have any effect on the migration of the nonhematopoietic melanoma cells, MEL1106. As can be seen in supplementary Fig. 4, while cytochalasin D inhibited cell migration, SMC #13 did not have any effect on cell movement relative to the vehicle-treated cells (Supplementary Fig. 4a-c).

**SMC #13 does not affect naïve healthy primary cells**. Since disassociation occurs between the C-terminal region of WIP and the N-terminal region of WASp, leading to the exposure of WASp degradation sites[40], it is likely that this dissociation upon activation is necessary to enable the binding of SMC #13. SMC#13 binding may thus affect the equilibrium of the WIP-WASp complex and skew it in favor of the unbound open conformation. To examine this possibility, we determined whether SMC #13 selectively affects activated hematopoietic cells, including malignant cells. For this purpose, PBMCs were isolated from the blood of healthy donors[56]. The PBMCs were either left untreated or were activated with PMA and ionomycin. Both naïve and activated cells were treated either with SMC #13 or with vehicle. The cellular concentration of WASp was evaluated using western blot. The normalized WASp levels of the naïve and activated SMC #13-treated cells were calculated relative to vehicle-treated cells (Fig. 4a, Supplementary Fig. 11). Downregulation of WASp levels

($72.8 \pm 7.3\%$) was observed only in activated SMC #13-treated PBMCs but not in naïve PBMCs obtained from healthy donors ($P \leq 0.04$).

To further investigate the effect of SMC #13 on naïve vs. activated PBMCs, we determined the proliferative capacity of these cells using an XTT-based proliferation assay. To this end, PBMCs were isolated from blood samples of healthy donors and were left inactivated, or activated with PMA and ionomycin, following treatment with SMC #13 or vehicle as control (Fig. 4b). Naïve PBMCs that were treated with SMC #13 did not show any significant change in proliferation. Upon activation, the proliferation rate of the PBMCs increased by $33.2 \pm 5.2\%$ relative to the naïve cells ($P \leq 0.02$). Strikingly, when activated PBMCs were incubated with SMC #13, their proliferation decreased by $73 \pm 10.2\%$ relative to the activated vehicle-treated PBMCs ($P \leq 0.01$). These data clearly suggest that SMC #13 selectively affects activated cells.

Next, we examined whether SMC #13 affects the migration of freshly isolated PBMCs from healthy donors. The cells were activated with PMA and ionomycin or left inactivated, followed by treatment with SMC #13 or vehicle. Remarkably, naïve PBMCs did not exhibit any significant change in their chemotaxis displacement when treated with SMC #13 versus vehicle. However, when the cells were preactivated and treated with SMC #13, their directional migration was impaired and their cell displacement was significantly decreased, to a distance of $36.4 \pm 2.7$ μm relative to untreated activated cells, which migrated to a distance of $76.1 \pm 3.2$ μm ($P \leq 0.00004$), (Fig. 4c and Supplementary movies 5-8). Collectively, these results demonstrate that SMC #13 affects only activated PBMCs. Furthermore, activated PBMCs treated with vehicle showed significantly higher WASp levels, which may be correlated with higher WASp-dependent cellular activity in malignant cells, in analogy to other WASp homologs e.g., N-WASp and WAVEs[88,89].

**SMC #13 lacks cytotoxic side effects**. To evaluate the potential safety of SMC #13, we examined the median toxic dose (TD$_{50}$) of SMC #13 by measuring cell viability using the propidium iodide (PI) assay (Supplementary Fig. 5). Freshly isolated PBMCs from a healthy donor were incubated with increasing concentrations of SMC #13 ranging from 18.75 to 600 μM. After incubation with SMC #13 for 24 h, the cell viability was determined relative to the negative control-treated cells. Using a linear trend line formula, the TD$_{50}$ was calculated as 171 μM.

Next, we evaluated the potential of SMC #13 to induce hemolysis. At the lowest tested concentration, 0.3125 mM of SMC #13, which is 7.8-fold higher than the concentrations we used in the in vitro assays (40 μM), a minor and insignificant level ($0.7 \pm 0.07\%$) of hemolysis was detected. Even at the maximal concentration of 5 mM, which is 125-fold higher than the in vitro concentration (40 μM) only $21.9 \pm 1.1\%$ hemolysis was observed (Supplementary Fig. 6a).

We further investigated the possible propensity of SMC#13 to provoke an immune response and cytokine secretion. Following incubation of SMC #13 with PBMCs from healthy donors, the secretion of key human inflammatory cytokines including IL-6, IL-10, TNF-α, and IFN-γ was determined using an ELISA-based cytokine assay. The concentrations of these inflammatory cytokines were not significantly different between SMC #13 incubated cells and the negative controls (Supplementary Fig. 6b). These findings demonstrate that the immune system is largely unaffected by SMC #13, and that this agent does not induce cytokine release.

**The safety of systemic SMC #13 administration in vivo**. To determine the suitability of SMC #13 as a lead compound for

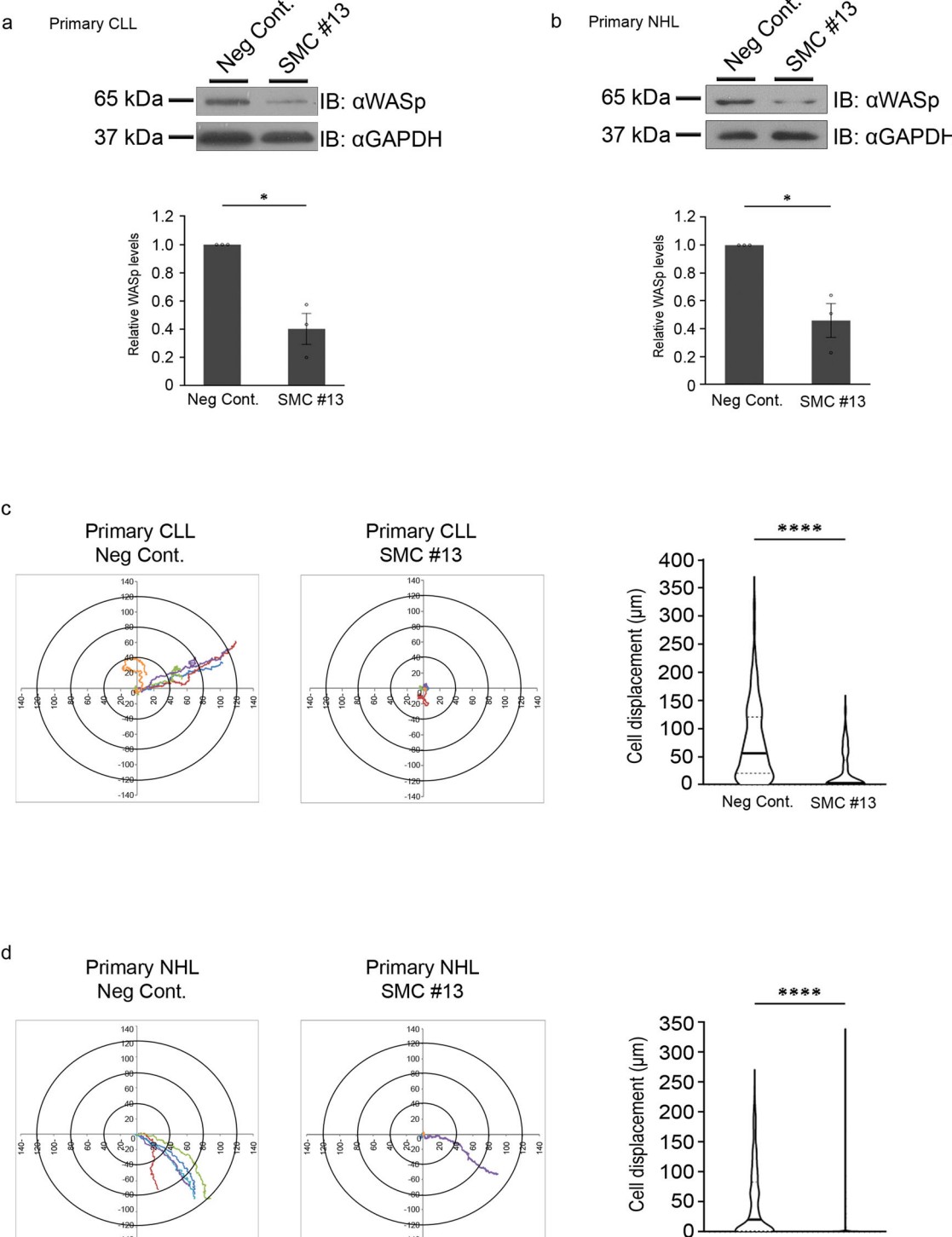

**Fig. 3 Downregulation of WASp cellular concentration by SMC #13 affects primary cancer cell activities. a** Effect on WASp levels: Primary CLL cells ($P = 0.0320$) or **b** primary NHL cells were incubated with 40 μM SMC #13 or vehicle for 24 h ($P = 0.0469$). Cells were lysed, and analyzed by immunoblot with anti-WASp and anti-GAPDH as a loading control. Densitometric analysis of the bands was performed with ImageJ, and is presented below the blots. Relative WASp levels were normalized to GAPDH and compared to the control. Values are shown as mean ± SEM of three independent experiments. The $p$ values were calculated using two-tailed paired $t$-test. **c, d** Migration assay following incubation with 40 μM SMC #13 in **c** primary CLL ($P < 0.0001$), or **d** primary NHL cells ($P < 0.0001$). Following 24 h incubation, cells were seeded on a chambered coverslip precoated with ICAM-1, and the random movement of the cells was assessed using Zeiss Observer Z1 inverted microscope and automatically tracked and analyzed. Left panel: Cell tracking analysis of five cells from a representative movie of each group. Each line represents the pathway of a single cell. Right panel: Graph summarizing the mean cell displacement (Neg Cont.: $n = 104$ CLL cells, 467 NHL cells; SMC #13 $n = 58$ CLL cells, 541 NHL cells). Values are shown as mean ± SEM; the $P$ value of two-tailed unpaired $t$-test is shown.

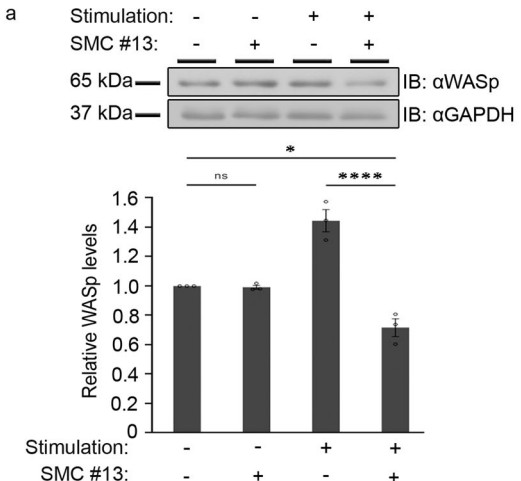

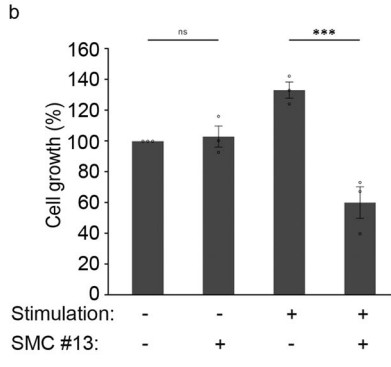

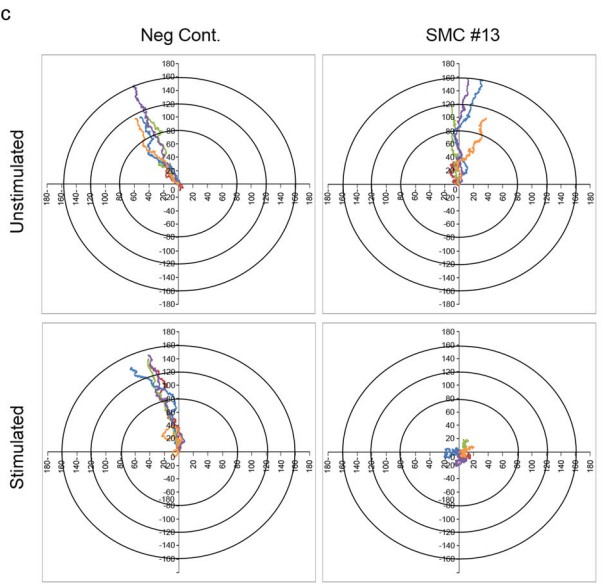

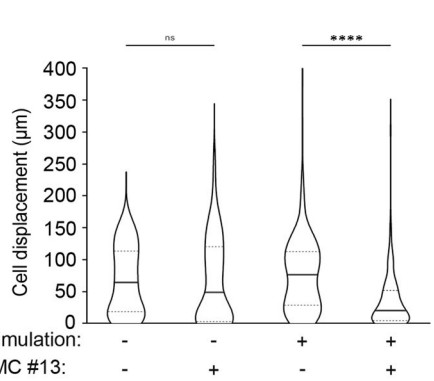

**Fig. 4 SMC #13 does not affect WASp cellular concentration and WASp-dependent cellular function in healthy naïve primary cells. a** PBMCs were incubated with 40 μM SMC #13 or vehicle, and were left unstimulated, or stimulated with 1 μg/ml ionomycin and 2.5 ng/ml PMA for 24 h (($P < 0.0001$) and ($P = 0.0140$ for vehicle unstimulated vs SMC#13 stimulated)). The cells were then lysed and analyzed by immunoblot with anti-WASp antibody and anti-GAPDH as a loading control. Densitometric analysis of the bands was performed with ImageJ, and the values are presented below the blots. WASp cellular concentration levels were normalized to GAPDH and compared relative to the unstimulated vehicle-treated cells. Values are shown as mean ± SEM of three independent experiments. The $P$ values were calculated using one-way anova and Tukey's multiple comparison test. n.s not significant. **b** XTT proliferation assay was performed following 40 μM SMC #13 incubation and/or stimulation of PBMCs, as described in **a**. The graph summarizes the mean percent cell proliferation relative to the control ($P = 0.0003$ for stimulated vehicle vs SMC#13). Values are shown as mean ± SEM of three independent experiments. The $P$ values were calculated using one-way anova and Tukey's multiple comparison test. n.s not significant. **c** PBMCs were incubated with 40 μM SMC #13 or control, left unstimulated or stimulated for 24 h, and then seeded on a chambered coverslip precoated with ICAM-1. Cell migration was assessed using Zeiss Observer Z1 inverted microscope. Left panel: Cell tracking analysis of five cells from a representative movie of each group. Each line represents the pathway of a single cell. The graph on the right summarizes the mean cell displacement (Neg Cont. (−): $n = 301$; SMC #13 (−): $n = 255$; Neg Cont. (+): $n = 301$; SMC #13 (+): $n = 264$) ($P < 0.0001$ for Stimulated vehicle control vs SMC#13). Values are shown as mean ± SEM of three independent experiments. The $P$ values were calculated using one-way anova and Tukey's multiple comparison test. n.s not significant.

in vivo studies, we examined whether it induced toxic effects in mice. Diabetic/severe combined immunodeficiency (SCID/NOD) mice received a single intraperitoneal (i.p) injection either with 100 mg/kg of SMC #13 or vehicle only as a control group. There was no evidence of weight loss (Supplementary Fig. 7a), lethargy, or other physical indicators of morbidity or stress over the course of 21 days following treatment. To further examine whether

SMC #13 could be tolerated as a drug used for extended treatment, we tested safety of 100 mg/kg in multiple i.p injections. (Supplementary Fig. 7b). No toxic effects or other indicators of morbidity or tissue damage to the heart, lung, liver, kidney, bowel, or spleen (macroscopic or microscopic) were detected. These studies support the safety of SMC #13 for in vivo pre-clinical efficacy studies.

**Durability in the plasma and pharmacokinetics of SMC #13 in vivo.** We next characterized the stability and distribution of SMC #13 in vivo, by measuring the pharmacokinetics in the plasma and within the tumors of mice bearing human Burkitt's lymphoma tumors (Raji), using High-Performance Liquid Chromatography (HPLC) for drug detection. The SCID/NOD mice were engrafted subcutaneously with Raji cells between their shoulders. Once the tumors developed and their volume reached an average of 300 mm$^3$, the mice were injected i.p with 100 mg/kg of SMC #13. Mice were sacrificed at the indicated timepoints, and samples from both blood and tumors were analyzed (Fig. 5a and Supplementary Fig. 8). The concentrations of SMC #13 at the various timepoints are shown in Fig. 5a. Notably, the $C_{max}$ in the tumor was $97.2 \pm 14.5\,\mu$M at the 30 min timepoint (Fig. 5a, red curve). These data suggest that SMC #13 can be administered systemically through the bloodstream, and that it reaches the in vitro therapeutic concentration of 40 $\mu$M inside the tumor within the first 30 min.

Furthermore, we performed a plasma protein binding (PPB) assay of SMC #13, evaluating its binding to full plasma proteins and in vitro cultured media supplemented with serum. Following 30 min incubation of SMC #13 with full plasma, medium, or PBS as a control, unbound SMC was separated from the protein bound SMC #13 complex using a molecular weight cutoff of 10kD and measured using HPLC. The calculated PPB of SMC #13 was measured to be ~85.9%, while in vitro media supplemented with fetal calf serum (FCS) yielded 78% binding (Supplementary Fig. 9). Furthermore, we also predicted the PPB value of SMC#13 to be 85% using "preADMET", a virtual predictor[90].

**SMC #13 suppresses in vivo tumor growth of human NHL cells following systemic administration.** To determine whether SMC #13 could suppress in vivo tumor growth, we determined the efficacy of SMC#13 in vivo on human NHL cells. We subcutaneously engrafted Raji cells in SCID mice. Once the tumors reached an average of ~100 mm$^3$ in volume, mice were randomly divided into two groups to receive i.p injections of 100 mg/kg SMC #13 or vehicle only every 72 h, for a total of six injections. Tumor volume was measured daily to determine the effect of SMC #13 on tumor growth (Fig. 5b) and on the average growth rate of the tumors from the first treatment (Fig. 5c). SMC #13-treated mice showed significantly smaller tumor volumes with an average of $457.7 \pm 83$ mm$^3$ and growth rate of $23.8 \pm 5.3$ mm$^3$/day compared to the vehicle-treated mice that presented an average tumor volume of $1826.1 \pm 233.7$ mm$^3$ and growth rate of $106.7 \pm 14$ mm$^3$/day ($P \leq 0.001$ and $P \leq 0.0001$) (Fig. 5b–d). In addition, the number of cells isolated from the tumor was significantly lower in SMC #13-treated mice compared to the control group ($P \leq 0.05$) (Fig. 5e). Furthermore, treatment with SMC #13 was effective at significantly prolonging survival in tumor-bearing mice (Fig. 5f).

**SMC #13 suppresses WASp-dependent cellular functions in human NHL tumor cells recovered following treatment.** Ex vivo analyses of WASp cellular concentration and its regulation on NHL tumor obtained samples was performed 24 h after the last i.p injection into mice. Consistent with the previous findings, WASp levels were decreased by $46.3 \pm 6.4\%$ in cells obtained from the SMC #13-treated mice compared to the control group, as detected by western blot ($P \leq 0.01$) (Fig. 6a, Supplementary Fig. 11). Cellular proliferation of SMC #13-treated tumor cells was decreased by $80 \pm 3\%$ ($P \leq 0.0001$) relative to cells from the control-treated mice (Fig. 6b). Integrin activation was decreased by $30.8 \pm 3.4\%$ in the treated group compared with the control group ($P \leq 0.02$) (Fig. 6c, Supplementary Fig. 12b).

Furthermore, systemic administration of SMC #13 led to a reduction in the ex vivo migratory capacity of the lymphoma cells, with an average displacement of $20.5 \pm 2.6\,\mu$m in comparison to $80.7 \pm 8.5\,\mu$m in the vehicle-treated cells ($P \leq 0.0001$). Downregulation of WASp via SMC #13 resulted in an average decrease of 74.6% in the migration of B-NHL cells (Fig. 6d). Next, we determined the invasive capacity of tumor cells by measuring ECM degradation. Equal numbers of tumor cells isolated from SMC #13 and vehicle-treated mice were incubated for a period of 24 h on fluorescent gelatin. The invasive capacity was measured by gelatin degradation, and was calculated relative to the cell area as determined by F-actin staining using phalloidin. Strikingly, tumor cells from control mice degraded an average gelatin area of $92.2 \pm 2.9\%$ compared to the SMC #13-treated tumor cells, which degraded an average gelatin area of $31.5 \pm 3.3\%$ ($P \leq 0.0001$) (Fig. 6e).

To further elucidate the impact of SMC #13 on the tumor cells, we examined the expression levels of the proliferation marker Ki-67, and Caspase-3 as an apoptosis marker, using immunohistochemistry (IHC) in mouse tumor sections. Histological analysis showed a significant difference in proliferation as measured by Ki-67 staining (Fig. 6f). Ki-67 was reduced by two fold in tumor sections obtained from SMC #13-treated mice relative to the control mice. The tumor sections were stained with an antibody that recognizes the active cleaved form of Caspase-3 to detect apoptotic cells. We observed a significant increase of $~2.45 \pm 0.01$ fold in apoptotic cells in sections from SMC #13-treated mice relative to control xenografts (Fig. 6g).

Taken together, our data demonstrate that SMC #13 specifically binds to the key regulator of the actin-cytoskeleton, WASp, enhances its degradation, and downregulates actin-based processes including integrin activation, cellular proliferation, migration, and invasiveness. This study demonstrates that this WASp-targeting SMC has powerful suppressive effects on human leukemia/ lymphoma cells and could potentially warrant translation towards testing in clinical trials.

## Discussion

Current treatments for certain types of hematopoietic malignancies have significantly improved overall survival (e.g., ALL, aggressive NHL), while available treatments for others (e.g., AML) are far from adequate. Furthermore, considerable challenges remain with standard chemotherapy, including lack of specificity, high toxicity, and multiple side effects[91–96]. Therefore, new treatment strategies are required to improve efficacy and reduce current drawbacks either as alternative monotherapies or combinatorial treatments.

To date, the role of WASp in malignant cells is poorly understood in comparison to the well-known properties of its homologs N-WASp and WAVE in non-hematopoietic cells. In this study, we leveraged the natural mechanism of WASp ubiquitylation and degradation as a strategy to specifically target hematopoietic cancer cells. The critical role of WASp as a key regulator of the actin cytoskeleton is well documented in WAS/XLT patients. Cells obtained from these patients exhibit poor activation, proliferation, and migration capabilities[71,73]. These actin-dependent processes[21] are hallmarks of cancer development[18]. Therefore, blocking WASp signaling and activity can be expected to disrupt these functions in hematopoietic malignancies. To develop such an inhibitor, we first utilized in silico screening with a machine learning platform and identified SMC #13 as a potential inducer of WASp degradation, thereby blocking WASp-dependent cytoskeletal dynamics in hematological cancers.

Since WASp stability highly depends on WIP, it has not been fully purified to date[97]. Thus traditional binding methods such as

IP Experiments

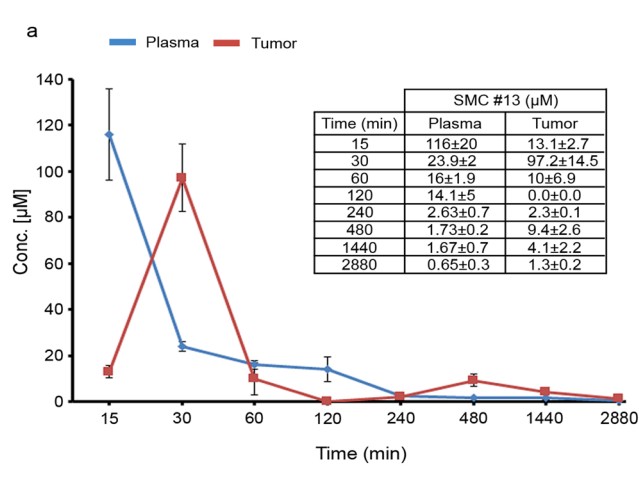

**a**

| Time (min) | SMC #13 (µM) | |
| --- | --- | --- |
| | Plasma | Tumor |
| 15 | 116±20 | 13.1±2.7 |
| 30 | 23.9±2 | 97.2±14.5 |
| 60 | 16±1.9 | 10±6.9 |
| 120 | 14.1±5 | 0.0±0.0 |
| 240 | 2.63±0.7 | 2.3±0.1 |
| 480 | 1.73±0.2 | 9.4±2.6 |
| 1440 | 1.67±0.7 | 4.1±2.2 |
| 2880 | 0.65±0.3 | 1.3±0.2 |

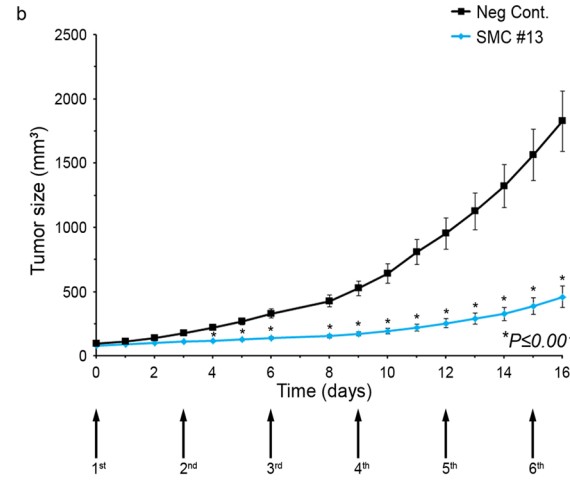

**b**

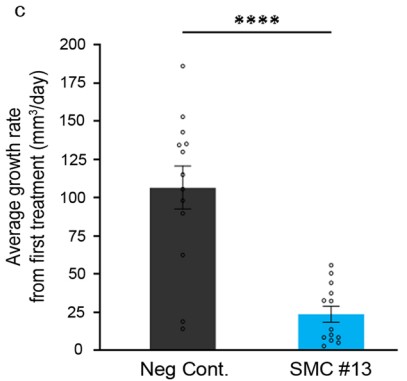

**c**

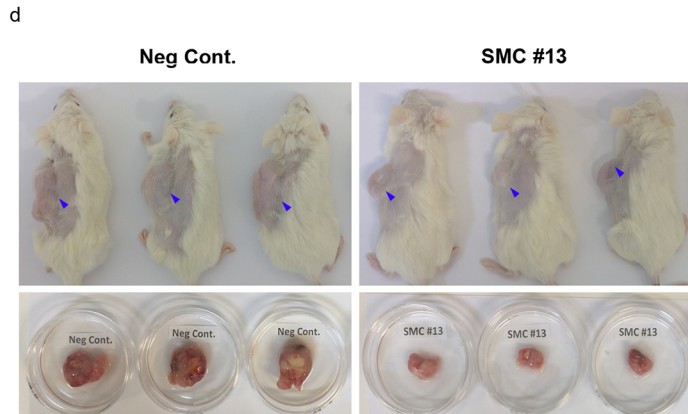

**d**

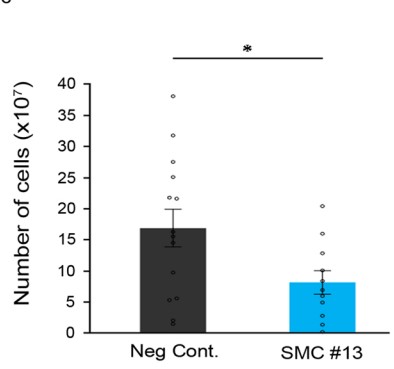

**e**

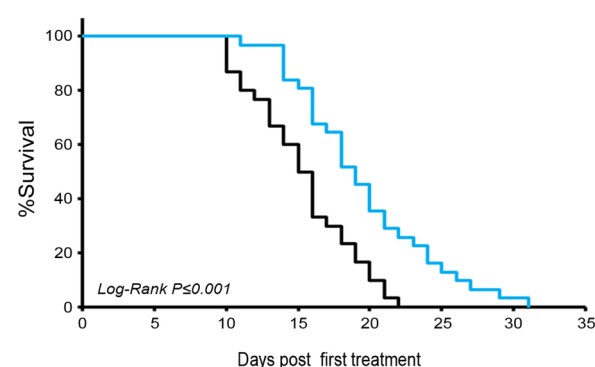

**f**

isothermal titration calorimetry (ITC) or surface plasmon resonance (SPR) are not feasible for working with the full length protein, and are challenging for use on fragments[98]. Thus, we determined SMC#13 binding to fluorescently tagged WASp by MST and interferometry analyses.

The differences between the estimated affinity of SMC #13 to WASp and the effective concentration needed for the induction of WASp degradation (Kd of ~30.8 nM relative to 40 µM,

respectively) might be explained by the high lipophilicity and moderate solubility of the SMC with a logD~3.3 (Supplementary Fig. 1d)[99] or alternatively due to the strength of WIP; WASp interactions under physiological conditions, which may be disrupted only under higher concentrations of SMC #13.

WASp is exclusively expressed in all hematopoietic lineages, excluding red blood cells[100]. The tissue-specificity of WASp expression makes it an attractive target for hematopoietic cell

**Fig. 5 In vivo efficacy studies of SMC #13 following i.p. administration in mice engrafted with human NHL. a** Pharmacokinetics. HPLC analysis of samples following i.p. injections of SMC #13 into SCID/NOD mice. The mice were injected IP with 100 mg/kg SMC #13. Blood samples and tumors were obtained at the indicated timepoints, and SMC #13 was detected by HPLC. The graph presents SMC #13 concentration kinetics. Values are shown as mean ± SEM. **b** Tumor volume as was measured daily in SCID/NOD mice treated with 100 mg/kg SMC #13 ($n = 13$) or control ($n = 15$). Arrows represent days of treatment. ($P \leq 0.001$). Values are shown as mean ± SEM. The P values were calculated using two-tailed paired $t$-test for each individual point. **c** Average growth rate of the tumor from the first treatment to the end point (between days 0 and 16) ($P < 0.0001$, $n = 13$). Values are shown as mean ± SEM. The P values were calculated using two-tailed paired $t$-test. **d** Representative image of the mice (upper panel) and their tumors (lower panel) at the end of the experiment. **e** Number of cells extracted from the tumor. ($P = 0.0322$, Negative control vs SMC#13) Values are shown as mean ± SEM (DMSO, $n = 14$; SMC#13, $n = 11$). The P values were calculated using two-tailed unpaired $t$-test. **f** Kaplan–Meier survival curve of NRG mice transplanted with Raji NHL cells and treated with vehicle ($n = 30$) or SMC #13 ($n = 31$). Statistical analysis was performed using log-rank test ($P \leq 0.001$).

targeting, limiting potential off-target effects to bystander non-hematopoietic cells. Torres et al. previously showed that WASp maintains a dynamic equilibrium between its open and closed conformations[101]. When hematopoietic cells are in the resting state (unstimulated), the dominant cellular form of WASp is a closed conformation, which is auto-inhibited by intramolecular interaction between the GBD and the VCA domains[101]. We and others demonstrated that WIP serves as a chaperone of WASp, and this interaction protects WASp from degradation via ubiquitylation[40,62,102–104]. Using triple-color FRET technology, we previously showed that phosphorylation of WIP on the serine 488 residue by PKCθ releases its C-terminus from interaction with the N-terminus of WASp, exposing the ubiquitylation sites of WASp to the E3 ligases Cbl-b and c-Cbl[40]. Our proposed mechanism of action for SMC #13 involves shifting of the WIP-WASp-binding dynamics, thereby enhancing the degradation of WASp (Fig. 7).

Additional selectivity of our therapeutic strategy lies on the observation that malignant cells are frequently highly activated, since they often acquire somatic mutations inducing activating signaling pathways or disrupting feedback mechanisms[18,23,105]. Thus, highly activated cells are more susceptible to SMC #13 treatment than naïve cells. A relatively common mutation in malignant cells occurs in the Ras GTPase superfamily, inducing cellular activation[106–108]. Several members of the Rho family are frequently mutated in a large variety of cancers. Specifically, CDC42 and its GEF, VAV, which are involved in WASp activation, are frequently mutated in hematopoietic cancers[109], leading to constitutive activation of this signaling axis. WASp activation affects its association with WIP[40,41]. Supporting evidence of the effect of WASp activation on its association with WIP came from the findings of Murga-Zamalloa et al. in NPM-ALK positive anaplastic large cell lymphoma (ALCL) cells[110]. This study showed that phosphorylation of WASp on Y102 induces its activation, leading to WIP disassociation and WASp degradation. Interestingly, in ALCL, impairing WASp activity by knock down or expression of the WASp mutant (Y102F) impaired the malignant phenotype. Finally, a constitutively phosphorylated WASp mutant (Y102E) showed decreased association with WIP, which decreased WASp stability. These findings support the notion that interfering with the WIP-WASp interaction using SMCs may mitigate a variety of hematopoietic cancers by enhancing WASp degradation.

We searched the Cancer Genome Atlas (TCGA) database to evaluate the role of WASp in human patients with hematopoietic malignancies. For this purpose, we used the UALCAN portal[111] to correlate between the survival statistics of patients with hematopoietic malignancies and WASp or N-WASp expression (Supplementary Fig. 10). As can be seen in Fig. S9a, patients with lymphoblastic acute myeloid leukemia (LAML) who demonstrated high WASp expression experienced shorter lifespan relative to those exhibiting low/medium WASp expression ($P = 0.032$). In contrast, when the patient survival curves were plotted in correlation to N-WASp expression, no significant difference was documented between high or low/medium expression (Supplementary Fig. 10B). Similar correlations with WASp expression were seen in diffuse large B-cell lymphoma (DLBCL), but due to a small sample number of patients in the TCGA database, the differences were not significant (Supplementary Fig. 10c-d). The TCGA database includes only these two types of hematopoietic malignancy.

Contemporary therapeutic approaches mainly focus on upstream receptors or central aberrant signaling pathways. These approaches may be ineffective due to alternative pathways that may lead to primary, acquired, or adaptive resistance in patients, causing relapse. However, the direct degradation of WASp, a key global effector protein, should remain a sensitive target regardless of the mutation profile of a specific lymphatic cancer or patient. The activity of WASp is required to enable cellular functions of the cancer cells, and thus, disrupting WASp function is expected to result in a potent antitumoral effect even in cells without a specific mutation.

To conclude, SMC may serve as a prototypic strategy for targeting WASp and additional hematopoietic-specific cytoskeleton proteins. The demonstrated safety and efficacy of WASp-targeting SMC in vivo, together with the advantages inherent to the strategy of disrupting a key cellular effector protein of lymphatic cancer cells highlights the great promise of this therapy, and the importance of its further development toward clinical studies.

## Methods

**Reagents**. Antibodies were obtained from the indicated suppliers. The following primary antibodies were used for western blotting: mouse anti-GFP 1:500 (Roche Applied Science—Catalogue no. 11 814 460 001), mouse anti-WASp D1 1:250 (sc-5300) and rabbit anti-GAPDH 1:1000 (Santa Cruz—sc-47724). HRP-conjugated goat anti-mouse 1:10,000 (Jackson—115-035-003) was used as a secondary antibody. Proteasome activity was blocked by addition of MG132 (Sigma–Aldrich—C2211) at a final concentration of 50 μM for 3 h before the cells were harvested. The following primary antibodies were used for flow cytometry: mouse anti-WASp D1 1:250 (Santa Cruz—sc-5300). The KIM127 hybridoma was purchased from the ATCC (KIM127 (ATCC® CRL2838™)) and the supernatant was purified to obtain the antiactivated human LFA-1 antibody (Sigma–Aldrich). For flow cytometry, secondary Alexa Fluor 488-conjugated goat anti-mouse IgG1 antibody was used 1:2500 (Jackson – A11034). SMC #13 was synthesized and purified (>90% purity) using LC-MS/MS by ChemBridge Research Laboratories LLC.

**Protein modeling**. A 3D model (PDB ID: 2IFS) of the WH1 domain of N-WASp[45] was downloaded from the Protein Data Bank (PDB)[112]. Using Discovery Studio 3.0 (BIOVIA) Virtual mutagenesis tool, the model was adapted to the sequence of the WASp WH1 domain.

Specifically, the following changes were introduced: V71A; A72V; L74F; R81K (positions relate to the sequence of human WASp).

Additionally, the WIP peptide and a portion the protein linker were deleted from the model.

**Candidate library preparation**. Two libraries of available drug-like small molecules were obtained from ChemBridge Corporation (San Diego, California, USA) and from the NIH MLSMR in the SDF (structure-data file) format. Using the FAST

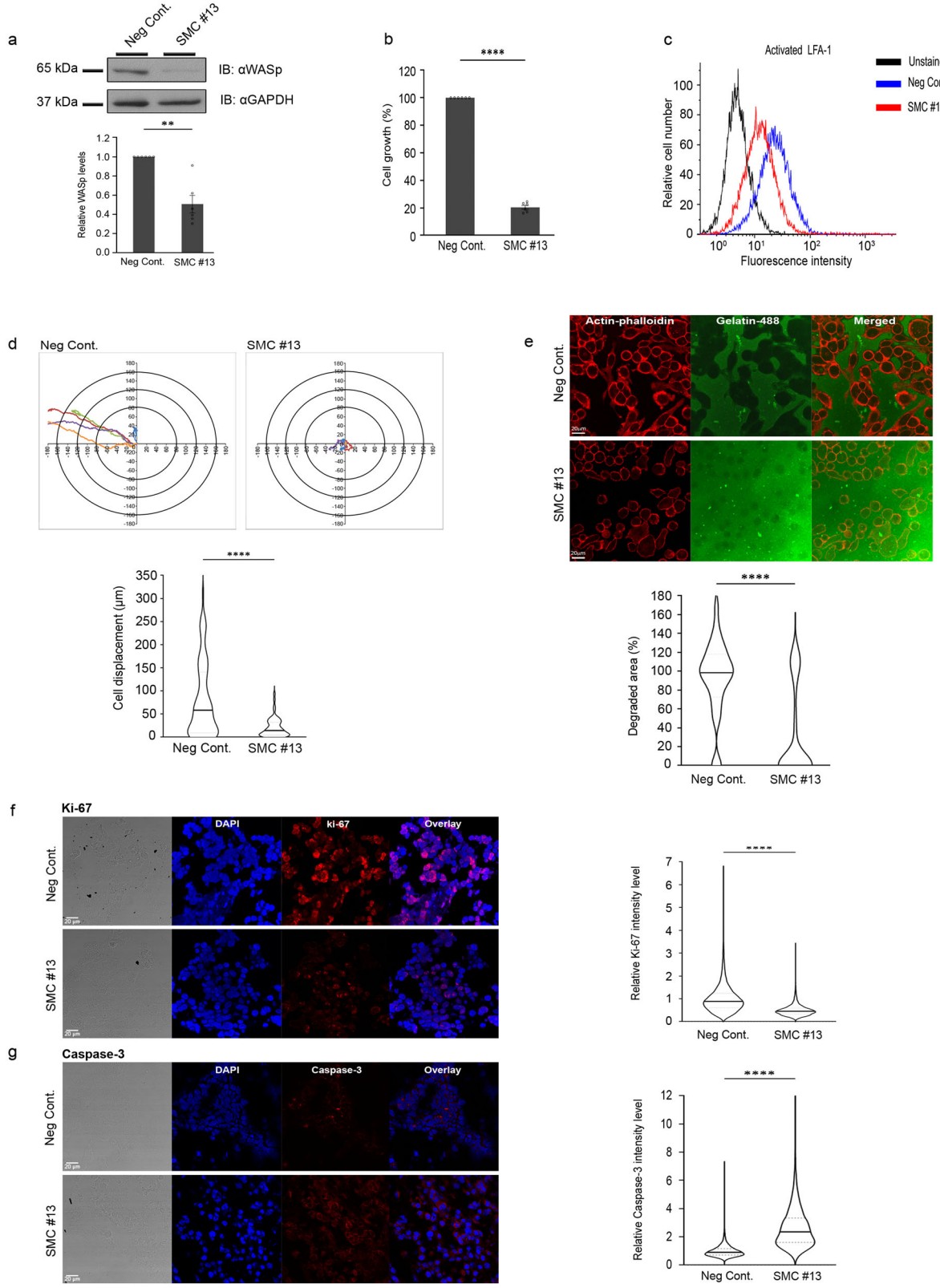

algorithm[47,48] as implemented in Discovery Studio 3.0, from each file, up to ten possible 3D conformations were generated and saved in the PDB format.

**Virtual docking and screening**. Virtual screening was performed using a machine learning classifier trained to distinguish between docking models of experimentally validated protein-small molecules pairs (positive examples) and randomly paired protein and small molecules (negative examples), taken from a curated 3D protein/drug interaction dataset[113]. Docking (including redocking of

positive examples) was carried out with the use of PatchDock[49,50], with the settings of 'drug' and '4' as the molecule and cluster parameters (respectively). The Random Forest algorithm[114] as implemented in R[115] was used for training the classifier, with the tree number parameter set to 500.

Discovery of WASp-binding candidates was carried out using a set of commercially available small molecules collection NIH's MLSMR library and from ChemBridge Corporation's CORE Library compound collection. For each small molecule in this set, we performed virtual docking to the modified 2IFS model.

**Fig. 6 SMC #13 inhibits WASp-dependent cellular functions ex vivo in cells obtained from tumors of mice following i.p. administration. a** WASp levels as detected by western blot analysis. Cells obtained from tumors of SMC #13- or control-treated mice were lysed, and WASp expression was analyzed by immunoblot, and normalized to GAPDH as a loading control. Densitometric analysis of the bands was performed with ImageJ. Relative WASp levels were compared to vehicle-treated mice and are presented in the graph below ($n = 6$, $P = 0.0030$). Values are shown as mean ± SEM. The $p$ values were calculated using two-tailed paired $t$-test. **b** Cell growth analysis as detected by XTT-based proliferation assay of cells extracted from tumors of SMC #13- or vehicle-treated mice ($n = 6$, $P \leq 0.0001$). Values are shown as mean ± SEM. The $P$ values were calculated using two-tailed paired $t$-test. **c** Surface staining of the active-LFA-1 on tumor cells of SMC #13- or vehicle-treated mice, as analyzed by FACS ($n = 8$). **d** Cells extracted from tumors of SMC #13- or vehicle-treated mice were seeded on a chambered coverslip precoated with ICAM-1. Cell migration was assessed using Zeiss Observer Z1 inverted microscope. Left panel: Cell tracking analysis of five cells from a representative movie of each group. Each line represents the pathway of a single cell. The graph on the right summarizes the mean cell displacement. (Neg Cont.: $n = 87$; SMC #13: $n = 58$). ($P \leq 0.0001$). Values are shown as mean ± SEM. The $P$ values were calculated using two-tailed unpaired $t$-test. **e** Cells extracted from SMC #13- or vehicle-treated mice were incubated overnight over fluorescent gelatin coverslips. Following incubation, cells were fixed, permeabilized and counterstained with phalloidin 594. Images were acquired with a Zeiss Confocal Laser Scanning Microscope (LSM 510 META). The images were analyzed using ImageJ software, and the percentage degradation was determined by the equation: [(degraded area)/(cell area)]×100. (Neg. Cont.: $n = 31$ cells; SMC #13 $n = 46$ cells; 3 independent experiments). ($P \leq 0.0001$). Values are shown as mean ± SEM. The $p$ values were calculated using two-tailed unpaired $t$-test. **f** Tumor sections from SMC #13- or control- treated mice (i.p. injections) were stained with DAPI and with anti-Ki-67 Ab followed by Alexa Fluor 594-conjugated goat anti-rabbit secondary Ab ($P \leq 0.0001$), or with **g** anti-cleaved Caspase-3 Ab followed by Alexa Fluor 594-conjugated goat anti-rabbit secondary Ab. Bright-field and fluorescence images were acquired with a Leica Confocal Laser Scanning Microscope (Leica SP8). ($P \leq 0.0001$). Values are shown as mean ± SEM. The $p$ values were calculated using two-tailed unpaired $t$-test.

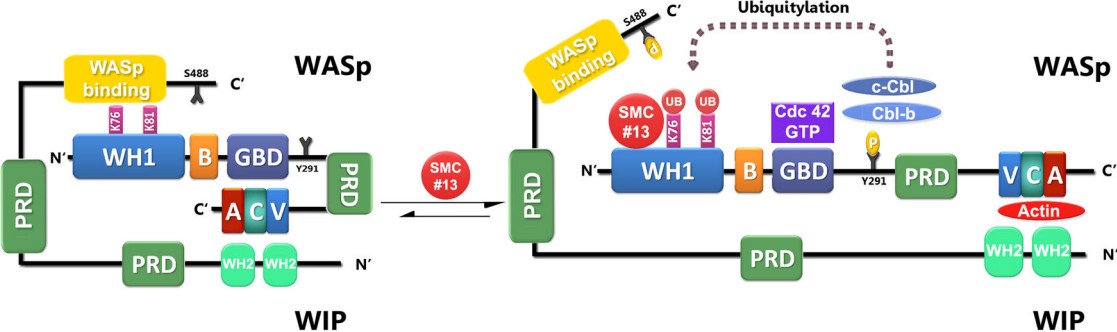

**Fig. 7 A suggested model for WASp downregulation by SMC #13.** Following cellular activation, WASp undergoes ubiquitylation on lysine residues 76 and 81, located at the WH1 domain, directing WASp to proteasomal degradation. WIP masks WASp ubiquitylation sites, thereby protecting it from degradation. Upon cellular activation, WASp is released from its auto-inhibitory state and results in a partial dissociation from the C'-terminus of WIP at the N'-terminal WH1 domain of WASp, while retaining its interaction between the C' terminus of WASp and the N'-terminus of WIP. As a result, WASp is subsequently degraded by the E3 ubiquitin ligases, the Cbls. SMC #13 enhances WASp degradation in activated cells by further interfering with the WIP;WASp interaction, thereby enhancing WASp ubiquitylation.

Then, the results of the PatchDock docking were scored by the trained Random Forest model to predict whether or not the docked models represent pairs that are likely to interact. Then, the small molecules that received the highest score were manually inspected to determine which of them should be purchased for experimental assessment.

**Model redocking and verification.** Forcefield-based optimization was performed for the selected molecules via the Discovery Studio 3.0 'Clean Geometry' tool. Noncovalent bonds between the SMC and the protein were computed with Discovery Studio 'Monitor' tool. The resulting putative models were then visually inspected in comparison to the pocket containing ubiquitylation targeted lysine residues 76 and 81 (calculation of the pocket was performed with the use of 'Pharmacophore Modelling' suite in Discovery Studio). Putative models were then inspected for steric clashes between the SMC and the WIP portion of the 2IFS model, using the Discovery Studio 'Monitor' tool.

**Cell lines and primary cells.** Blood samples from patients were collected at the Sheba Medical Center with written informed consent in accordance with the declaration of Helsinki, and the study approval by the Bar-Ilan University Ethics Committee. Primary cells, Human Embryonic Kidney (HEK) 293 T cells, Jurkat (clone E6.1) T-ALL and Raji Burkitt's lymphoma were cultured in RPMI 1640 (Sigma–Aldrich) supplemented with 10% FBS (Biological industries), 2 mM L-glutamine, 50 mg/ml penicillin, 50 mg/ml streptomycin (Sigma–Aldrich). All cells were grown at 37 °C, under an atmosphere containing 5% $CO_2$.

**Primary PBMC isolation.** Human primary PBMCs were isolated from whole blood of healthy donors or patients, as previously described[56]. Briefly, mononuclear cells were isolated by Ficoll density gradient (Axis Shield—1114547-1) centrifugation. Cells were activated using PMA (Sigma–Aldrich P1585) (2.5 ng/ml) and Iono-mycin (Merck Millipore—407950) (1 µg/ml) or left inactivated in the presence or absence of SMC#13 for 24 h.

**WASp knockout cells using CRISPR/CAS9.** CRISPR/CAS9 knockdown of endogenous WASp in T cells was conducted according to published protocol[116]. The pSpCas9 (BB)-2A-GFP (PX458) vector was purchased from Addgene, plasmid # 48138. The following RNA guides were designed and subcloned into pSpCas9 (BB)-2A-GFP.

5' aaacTCGCTGGAGATGTAAGTGGATc 3'
5' caccgATCACTTACATCTCCAGCGA 3'

**MST measurements.** YFP-tagged WASp, WAVE2-, and N-WASp were purified from cell lysates of HEK 293 T cells, transiently transfected with 10 µg of YFP-WASp, YFP-WAVE2, and N-WASp-YFP by DNA-calcium phosphate co-precipitation[40]. Forty-eight hours post transfection, cells were lysed using ice-cold lysis buffer consisting of 50 mM Tris- HCl (pH 7.6), 150 mM NaCl, 5 mM EDTA, 5 mM EGTA, 50 mM NaF, 1% NP-40, and complete protease inhibitor tablets (Roche). MST measurements were performed using the protein purification-free method described by Khavrutskii et al.[54]. Briefly, SMC#13 or Y-27632, a ROCK1 inhibitor, were diluted to a stock concentration of 400 µM, and serially diluted (200 µM to 1.5 nM) in 200 µL PCR-tubes. Then, the cell lysate was added to the

tubes at a 1:1 volume ratio, and the samples were gently mixed. The samples were incubated at room temperature for 30 min before being loaded into standard-treated Monolith™ capillaries (NanoTemper). After loading into the instrument (Monolith NT.115, NanoTemper), microscale thermophoresis was carried out at 50% blue LED power and 60% MST power. The ratios between normalized initial fluorescence and post temperature-Jump fluorescence were calculated were plotted against the concentration of the serially diluted molecule. KD values were determined using the NanoTemper analysis software (MO.Affinity Analysis v2.1.3).

**BLI analysis**. The interaction between the small molecule and the YFP-tagged proteins expressed in HEK 293 cells was determined by biolayer interferometry using the Octet Red 96 instrument (Forte-Bio Inc. Menlo Park, CA). Super Streptavidin biosensor tips were prewetted with PBS to establish a baseline before antibody immobilization. Then biotinylated anti-YFP nanobody (ChromoTek GFP-Trap®—gtb-250 121150) was immobilized onto Super Streptavidin biosensors and the unbound streptavidin residues were blocked using biocytin (10 μg/ml). Next, the desired YFP-tagged proteins in whole-cell lysates were also immobilized. All the binding data were collected at 30 °C. The association experiments were conducted using cycles of equilibrium (10 min), baseline (1 min), association (2 min), and dissociation (5 min). Binding to SMC #13 was conducted using six different serially diluted concentrations ranging from 50 to 200 μM. The association and dissociation plot and kinetic constants were obtained with Forte-Bio data analysis software. Equilibrium dissociation constants (Kd) were calculated according to the dose-dependent binding response curve.

**Western blot analysis and immunoprecipitation**. For analysis of whole-cell lysates (WCL) or IP experiments, $3 \times 10^5$ cells or $10 \times 10^6$ cells were used, respectively[40]. Cells were lysed using ice-cold lysis buffer containing 1% Brij, 1% n-Octyl-b-D-glucoside, 50 mM Tris–HCl, pH 7.6, 150 mM NaCl, 5 mM ethylene-diaminetetraacetic acid (EDTA), 1 mM Na3VO4 and complete protease inhibitor tablets (Roche). Protein samples were resolved with sodium dodecyl sulfate-polyacrylamide gel electrophoresis (SDS-PAGE), transferred to nitrocellulose membrane, and immunoblotted with appropriate primary antibodies. Protein A/G plus-Agarose beads (Santa Cruz Biotechnology—sc-2003) were used for IP. A preclearing step of cell lysate with protein A/G was performed prior to immuno-precipitation with mouse anti-WASp H-250 (Santa Cruz—sc-8353) antibody.

**Cycloheximide chase assay**. WASp protein stability was determined in the presence or absence of the protein biosynthesis inhibitor, cycloheximide (CAY—14126-1). T-ALL cells were incubated with either SMC #13 or vehicle. After 12 h, fresh medium containing cycloheximide (0.1 mg/ml) was added. Cells were lysed at the indicated timepoints (0, 2, 4, 6, and 8 h).

**FRET analysis**. Fluorescence resonance energy transfer (FRET) efficiency and correction were measured by the donor-sensitized acceptor fluorescence technique[41,56–61]. Briefly, three sets of filters were used: one optimized for donor fluorescence (excitation, 458 nm, and emission, 465–510 nm), a second for acceptor fluorescence (excitation, 514 nm, and emission, 530–600 nm), and a third for FRET (excitation, 458 nm; emission, 530–600 nm).

**FRET correction**. FRET correction was performed as described in detail[41,56–61].

**FRET efficiency calculation**. The FRET efficiency (FRETeff) was calculated on a pixel-by pixel basis as described in detail[41,56–61] using the following equation: FRETeff = FRETcorr/ (FRETcorr + CFP) × 100%, where FRETcorr is the pixel intensity in the corrected FRET image, and CFP is the intensity of the corresponding pixel in the CFP channel image.

**Gelatin degradation assay**. Glass coverslips (10 mm) were placed in a 24-well plate and precoated with 0.01% Poly-L-lysine (Sigma–Aldrich). The coverslips were then coated with 0.1% Oregon green 488-conjugated gelatin (Invitrogen), cross-linked with 0.5% glutaraldehyde, and coated with 2 μg/ml SDF-1α (PeproTech—RKP48061). Lymphocytes were incubated overnight over the gelatin-coated coverslips. Following incubation, cells were fixed and counterstained with phalloidin 594. Images were acquired with the Zeiss Confocal Laser Scanning Microscope (LSM 510 META) under a ×63 objective lens. The images were analyzed using ImageJ software, and the percentage of degradation was determined for each cell according to the equation: %Degradation area = (Area of degraded gelatin/Area of corresponding cells) × 100.

**Proliferation assay**. Lymphocyte proliferation was assayed with an XTT-based Cell Proliferation Kit (Biological Industries Ltd.) according to the manufacturer's instructions. Briefly, $1 \times 10^6$ cells were preincubated with the relevant treatment. After 24 h, 100 μl of cell suspension was seeded in triplicate in 96-well microplates and incubated with an XTT substrate; the light absorption of the XTT reduction products (450 nm) was monitored over time using a microplate plate-reader. Nonspecific reference absorption (630 nm) was measured as a control. Proliferation

was calculated by the differences in cell number 24 h after treatment relative to the control.

**Toxicity assay**. Healthy donor PBMCs ($1 \times 10^6$ cells/ml) were incubated for 24 h with different concentrations of SMC #13 ranging from 18.75-600uM in the presence of 1.25% DMSO at each concentration and in the negative control. Toxicity was evaluated using Propidium iodide (PI) staining. For PI staining, after incubation with SMC #13, cells were centrifuged for 7 min at $400 \times g$ and resuspended in 500 μl staining buffer containing PI (0.5 ng/ml), followed by flow cytometry analysis using a FACSAria™ III (BD Biosciences ™) cell analyzer.

**Lymphocyte migration assay**. Migration assay was performed using $1 \times 10^5$ cells seeded over chambered coverslips (LabTek) precoated with 6 μg/ml ICAM-1-Fc (BioLegend) and 2 μg/ml SDF-1α (Reprokine) suspended in 20 mM Tris pH 9, 159 mM NaCl, and 2 mM MgCl$_2$ at 4 °C, overnight[85]. The chambered coverslips were preincubated with imaging buffer containing 10% FCS for blocking. Cells were incubated at 37 °C, 5% CO$_2$ for 30 min to allow them settle before imaging. The DIC images were acquired with the Zeiss Observer Z1 inverted microscope every 5 sec for 20 min under an ×20 objective lens using Zen software. The random displacement of the cells was automatically tracked and analyzed using the ImageJ software TrackMate plugin[87]. All the movies were taken using the same settings. An individual trace was assigned to each of the analyzed cells. One representative field out of >6 random fields is presented. Playback speed is fifteen frames per sec.

**Flow cytometry analysis**

*Extracellular staining*. Lymphocytes ($3 \times 10^5$) were stained with 1 μg/ml antiactive human LFA-1 antibody (purified from KIM127 hybridoma sup) for 30 min at 37 °C, followed by staining with Alexa Fluor 488-conjugated goat anti-mouse IgG1 secondary antibody (Jackson), 30 min on ice.

*Intracellular staining*. Cells ($1 \times 10^6$/ml) were either stimulated with 2.5 ng/ml PMA and 1 μg/ml ionomycin or left unstimulated. Cells were collected ($5 \times 10^5$ cells per sample), washed twice with PBS, and fixed with 0.4 ml 3.7% paraformaldehyde for 20 min at room temperature. After two additional washes, cells were permeabilized with 0.1% Triton X-100 in PBS for 4 min at room temperature. Cells were blocked for 45 min with 2% goat serum, followed by 45 min incubation with 0.1 μg mouse anti-WASp Ab, on ice. The samples were washed twice, and then incubated with 1:2000-diluted Alexa Fluor 488-conjugated goat anti-mouse IgG1 antibody (Jackson), for 45 min on ice. Fluorescence was analyzed by flow cytometry on a FACS Gallios system (Beckman Coulter).

**Cytokine release assay**. Freshly isolated PBMCs from healthy donors ($1 \times 10^6$ cells/ml) were incubated for 24 h with either vehicle only (DMSO) as control, 40 μM of SMC #13, or left untreated. For positive control, cells were incubated with 7 μg/ml PHA for 96 h After incubation, the supernatant from each sample was collected. The levels of human cytokines, IL-6, IL-10, TNF-α, and IFN-γ were determined using Human Mini ELISA Development kits (PeproTech).

**Scratch-wound assay**. Melanoma cells (MEL1106) were seeded at a concentration of $1 \times 10^4$ cells per well over culture-insert four-well in μ-Dish 35 mm (Ibidi®) with a silicon insert to create 500 μm cell free gaps. Once cells reached high confluency, they were treated with SMC #13 for 24 h or vehicle only as a negative control. For a positive control, cells were treated for 10 minutes with 0.5 μM cytochalasin D, which inhibits actin-dependent cell migration. To follow wound colonization which reflects a combination of both cell proliferation and cell migration over time, images were taken every 30 min for a total of 18 h of incubation at 37 °C with 5% CO$_2$. Five fields were captured in every experiment using DMi8 Leica wide-field inverted microscope (Leica Microsystems, Mannheim, Germany), equipped with an incubation chamber and sCMOS DFC9000GT Leica camera, and driven by LasX software, using a 10 × 1.4NA objective. The migratory capability was quantified using the ImageJ plugin[117] "Wound healing size tool", which allows quantification of the wound area, wound coverage of the total area, and average wound width.

**Mice**. SCID/NOD mice were purchased from the Jackson Labs. All mice used were from colonies that were inbred and maintained under SPF conditions at the Bar-Ilan animal house. Housing and breeding of mice and experimental procedures were done according to the Bar-Ilan University Ethics Committee guidelines (IRB: 82-10-18), and mice were euthanized using CO$_2$. For in vivo efficacy studies, 6–8-week-old male SCID/NOD mice were used. Mice were subcutaneously injected with $3 \times 10^6$ human Raji B-NHL cells, in PBS mixed with Matrigel (CORNING) (1:1 v/v ratio). Tumor size was measured daily using a digital caliper, and the following formula was used to calculate the tumor volume, Volume = 0.5 × (major axis) × (minor axis)$^2$. When the tumors reached 100 mm$^3$, mice were randomly divided into two groups, and were treated intratumorally (i.t.) or intraperitoneally (i.p.) every 3 days for a total of six treatments with 20 mg/kg of SMC #13 suspended in Tyrode's isotonic buffer with 20% cyclodextrin (i.t.), or 100 mg/kg SMC #13 dissolved in DMSO, or with vehicle (i.p.).

**Immunohistochemistry (IHC)**. Tumor tissues were harvested from mice and were cut into 5 mm-thick tissue blocks. The tissue blocks were covered with cryo-embedding media-O.C.T, and were frozen with 2-methyl-butane cooled by liquid nitrogen. The frozen tissue blocks were sliced into 5 μm sections and mounted on glass slides (SuperFrost Plus, Thermo Scientific). Sections were fixed in cold acetone for 10 min, dried, incubated in blocking solution [10% FCS, 1% BSA dissolved in 0.01 M phosphate-buffered saline (PBS), pH 7.2] for 30 min, and then incubated overnight at 4 °C with mouse anti-WASp 2 μg/ml (Santa Cruz), rabbit anti-Ki-67 6.5 μg/ml (Abcam—ab15580), or rabbit anticleaved Caspase-3 (Cell Signaling—CST-9664T) dissolved in PBS with 0.5% BSA. The tissue sections were rinsed twice in PBS and once in PBS with 0.1% TWEEN, followed by incubation with secondary antibodies (Alexa Fluor 568-conjugated goat anti-mouse, or Alexa Fluor 594-conjugated goat anti-rabbit) in PBS with 0.5% BSA for 45 min at R.T. Tissue sections were then rinsed twice in PBS and once in PBS with 0.1% TWEEN, followed by incubation with 0.3 μM DAPI (Invitrogen) in PBS for 15 min at R.T. Finally, the sections were rinsed twice in PBS, and mounted with mounting medium. Images were acquired with the Leica SP8 confocal inverted microscope under a ×63 objective lens.

**Pharmacokinetics**. Male 6–8-week-old SCID/NOD mice were subcutaneously injected with $3 \times 10^6$ human Raji B-NHL cells, in PBS mixed with Matrigel (CORNING—FAL356234) (1:1 v/v ratio). Tumor size was measured daily. When the tumors reached 400 mm$^3$, mice were treated i.p. with 100 mg/kg SMC #13. Mice were sacrificed using $CO_2$ at the indicated timepoints (0, 15, 30, 60, 120, 240, 480, 1440, and 2880 min) post injection. Tumors were extracted, and blood samples were collected using cardiac puncture into Eppendorf tubes containing 7.5 U Heparin. Plasma was separated using centrifugation at $2000 \times g$ and the supernatant was transferred and stored at −80 °C. Tumors were frozen under liquid nitrogen and stored at −80 °C. All samples were supplemented with 0.1 M NaOH, and SMC #13 was extracted using 2 ml ethyl acetate as follows: Samples were vortexed for 2 min. The tubes were centrifuged at $2000 \times g$ for 10 min. The resulting organic layer was separated and evaporated using Savant Speed VAC SC110A Concentrator Centrifuge.

**HPLC protocol**. The concentration of SMC #13 was measured using Hitachi Elite LaChrom HPLC System on a Hypersil GOLD 5 μm column (250 mm ID × 2.1 mm OD). The mobile phase consisted of a two buffer gradient system (Buffer A: H$_2$O:ACN:TFA 95:5:0.1; Buffer B: H$_2$O:ACN:TFA 10:90:0.1). Samples were reconstituted in a volume of Buffer B equal to that of the original sample, without Trifluoroacetic acid (TFA). Samples were loaded with an injection volume of 10 μl, and analyzed using the following linear gradient: 0–18 min—30–100% buffer B, 18–19 min—100–30% buffer B, 19–25 min—30% buffer B at a flow rate of 0.5 ml/min for the PK and 0.35 ml/min for the PPB with a temperature of 30 °C. SMC#13 was detected at 270 nm with a retention time of 9.3 and 13 min (respectively to the flow rate) using spike and quantity estimated from the calibration curve of standards prepared at 100, 25, 6.25, 1.56-, and 0.39 μM.

**Statistical analyses**. Data calculations were conducted in Microsoft Excel while data were graphed and statistical analysis was performed using Prism (GraphPad, v9). P values were calculated using a two-tailed paired or unpaired t-test. Where >2 conditions were compared, a one-way ANOVA with a Tukey post-hoc test or Dunnett's multiple comparison was used to calculate P values. In all cases, the threshold P value required for significance was 0.05. Survival analysis was performed using log-rank test by R software.

**Reporting summary**. Further information on research design is available in the Nature Research Reporting Summary linked to this article.

## Data availability

All experimental data regarding the in vitro and in vivo systems and specific binding of SMC #13 are provided in the text and in the source data file. Source data are provided with this paper.

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

## Acknowledgements
We thank Matan Y. Avivi, Anat Raiff, and Dr. Alex Vorvak for their technical assistance, Dr. Lisette Bronswijk-Deddens (Sartorius) for her help with the BLI data analysis and Prof. Bilha Fischer for her advice on the chemical characterization of the SMC. This research was funded in part by the Israel Science Foundation (grant no. 491/10), and in part by the Israel Innovation Authority.

## Author contributions
M.B.S., G.B. and N.J. designed the research. G.B., A.B.S., E.N., N.J., A.P., N.R., O.L., I.L., A.F., Y.O. and S.F. performed the experiments. M.K. and A.A. provided the blood samples of the patients with informed consent in accordance with the declaration of Helsinki (0196-13-HMO) and approval of Bar-Ilan University ethics committee, and Sheba medical center. G.B., A.B.S., E.N., N.J., A.P. and M.B.S. analyzed the data. M.B.S. wrote the manuscript.

## Competing interests
"WASp-protecting small molecules, compositions, methods and uses thereof in the treatment of innate and acquired immune-related disorders or conditions" PCT/IB2014/061907, patent author: M.B.S. The remaining authors declare no competing interests.
