## [Peer Review File · Nature Communications]

Targeting WASp Provides a Therapeutic Approach for Hematopoietic MalignanciesREVIEWER COMMENTS

Reviewer #1 (Remarks to the Author):

In this manuscript, the authors describe the identification of a small molecule that binds to the actin nucleation promoting factor WASP and increases the rate at which WASP is degraded. The authors show that this small molecule decreases the steady state concentrations of WASP in malignant hematopoietic cells and decreases the proliferation and migration of these cells. They demonstrate that the small molecule does not have an influence on the proliferation or migration of naïve hematopoietic cells, suggesting that it may have low toxicity. Finally, the authors show that the small molecule can suppress tumor growth and increase survival times in mice models of cancer. The work is of high general interest because of the potential of the findings to impact human health. However, some aspects of the work are unconvincing, and a number of claims are made in the manuscript that are not supported by the data.

My primary criticisms of the work are the biochemical characterization of the small molecule and the claims about its mechanism of action.

The authors use MST to determine the binding affinity of the small molecule to WASP. The error in the average fluorescence change is very high in these experiments, and the values reported for the samples in which no binding is reported to occur (Fig 1b-d) show a wide range of change in fluorescence values at different ligand concentrations. That this range is the same as range of values for what is assumed to be bound and unbound WASP in Fig. 1a makes the binding data in Figure 1 unconvincing.

The authors claim that the small molecule specifically binds WASP, and imply that it interacts with the WH1 domain of WASP, but their data do not support this conclusion. Even if the MST data had less error, the results could reasonably be interpreted as the small molecule binding WASP indirectly through another protein because no experiments are done with purified WASP.

To make the biochemical data more convincing, the authors should provide stronger evidence of the interaction. While the authors mention in the discussion that MST using lysate was the only way to measure the interaction because of problems purifying full length WASP, multiple groups have been able to purify and study the WH1 domain in isolation. Given that small molecule is hypothesized to bind to WH1, the authors should be able to use another method (e.g, ITC) to more convincingly show the interaction (and measure the affinity) between the small molecule and the purified WH1 domain.

In addition, more information about the docking and the small molecule identification should be included in figure 1. For instance, the authors should show the theoretical structure of the docked small molecule bound to the WH1 domain. Based on this model, the authors should comment on why the small molecule doesn't bind to the WH1 of N-WASP in the discussion. In the results section the authors should mention that WAVE doesn't have a WH1 domain, so it is not surprising that the small molecule wouldn't bind to WAVE based on their mechanism (Figure 1C). The authors should also show the chemical structure of the small molecule in figure 1.

A key conclusion of the study is that the small molecule stimulates degradation of WASP by competing with WIP for WASP binding. To support this conclusion, the authors should use competition binding assays with purified components, as described above.

Please clarify in each figure legend whether the error bars represent standard error or standard deviation.

The description of the influence of the small molecule inhibitor on WASP is confusing because of the use of the phrase "WASP expression". The authors' use of this phrase instead of "the cellular concentration of WASP" seems in some places to be inconsistent with their results. For instance, on line 170 they write " . . . WASp expression was almost completely abrogated." To most readers, this

will probably be interpreted as meaning that either the transcription or translation of WASP was decreased, whereas the authors' data suggest that it is instead that the small molecule stimulates degradation of WASP to decrease its steady state cellular concentration.

The authors should address why the concentration of the small molecule required to influence WASP concentrations in cells is many orders of magnitude higher than the affinity of the molecule for WASP. In addition, the author should discuss whether or not the small molecule has properties consistent with a drug (e.g. logP, mw, number of h-bond donors and acceptors). In addition, while it is stated at least once in the text, each figure legend should mention the concentration of the small molecule used in the experiments.

The sentence on line 259 is unclear. What do the authors mean by "partial dissociation" of WIP and WASP?

For the data in figure 4A, the authors should report a non-normalized concentration of WASP, since comparing the concentrations between stimulated and unstimulated cells would be useful. Related to this, the data in figure 4D seem difficult to rationalize. Why is there little to no motility in the stimulated cells treated with the compound when the concentration of WASP in these cells looks very similar to the concentration of WASP in the unstimulated cells?

Much of the discussion is redundant, in my view. Specifically:

- the first paragraph in the discussion re-establishes the need to develop better treatments for hematopoietic malignancies. This is established in the introduction so it is unnecessary in the discussion.
- Much of the discussion is a recap of the results. The authors could make the discussion more succinct by not recapping the results except to the extent necessary to discuss implications of the results that were not mentioned in the results.

The text in the discussion describing the rationale for using MST in lysates is somewhat misleading, since, as pointed out above, experiments could be done with the isolated WH1 domain.

Lines 469-473 are also somewhat misleading, in my view. The statement that "malignant cells . . . mainly express the open, activated form of WASp while naïve cells dominantly express the closed-autoinhibited form" is referenced with a paper that is purely biochemical/in vitro and doesn't compare malignant versus naïve cells. The authors should clearly state here that this statement is their speculation. Further, in my view, it is inaccurate to say that cells "express" one form or the other. The autoinhibited versus activated states are in equilibrium with one another. It's possible that the open conformation is more populated in the malignant cells, but it doesn't make sense to me to say that the malignant cells "express the open, activated form".

Related to the above comment, the authors seem to be using the concept of WASP open and closing to indicate 1.) exposure of ubiquitylation sites on the WH1 domain and 2.) release of the VCA segment of WASP from the GBD of WASP. What is the evidence that these two molecular events occur in concert?

In figure 8, why are the WH2 domains of WIP depicted as binding to the VCA segment of WASP?

Reviewer #2 (Remarks to the Author):

Biber et al present the results of an in silico small molecule screen with in vivo pharmacokinetic data (Raji NHL cells) and modest efficacy using a compound putatively targeting the interaction of WASP and WIP to accelerate ubiquitination and degradation.

The in silico approach is to be commended, the drug target is attractive and the author's published expertise on protein-protein interactions and cell imaging is outstanding. The potential for disease(s)

impact is high, although is a little generic - which sub-class of hematological malignancy would most likely benefit? PK studies are show durable concentrations and show at least via IP route, micromolar concentrations can be achieved towards clinical application.

Statistical methods are appropriate; as a non-kinase, non-epigenetic target, the paper will definitely influence thinking in cancer therapy.

However, there are three current limitations with the paper that limit the scope and conclusions:-
1] Insufficient detail to convince the reviewer that the drug is actually on target and intracytoplasmic. Microscale thermophoresis can sometimes be limited by non-specific or off-target beyond putative interaction site. Non-specificity can be a concern when binding K_d 30 nM is 1300 x lower than in vitro concentration required to alter WASp protein expression; it is also a concern when FRET is completely abolished in Figure 2C. Thus crucial to make sure the molecule is not binding a YFP-WASp interface. Many drugs (as shown in Mc1-1 DUB or CDK7 screens for instance) can change degradation of proteins at high micromolar concentrations through kinases or unfolded response stress etc.

Thus evidence of further on target specific could be improved by at least one of the following:-

- (i) further detail and modelling of the in silico screen approach showing docking
- (ii) performance of MST with mutated WASp lacking residues adjacent to lysine 76 and 81 that are predicted to interact with the molecule
- (iii) performance of an orthogonal binding method such as BLI, SPR, ITC
- (iv) demonstration that covalently tagged SMC#13 can pull down WASp
- (v) demonstration that a modified version of SMC#13 (enantiomer or removal of benzyl-hydroxyl) does not bind WASp or result in WASp degradation
- (vi) demonstration that other actin-depolymerizing drugs do not change WASp protein degradation stability.
- (vii) demonstration that GFP- or CYP-WASp give same K_d

2] Despite a lengthy introduction, few primary samples are tested (1 primary NHL, 1 primary CLL) and only Raji in vivo and Jurkats in vitro. To make conclusions regarding hematological malignancies this is premature. Which NHL sub-type do the authors predict will be most sensitive? Follicular or Burkitt? T-ALL vs B-ALL? What about M7 AML?

3] Despite development of an invaluable tool for WASp-WIP interaction and modest in vivo, there is little mechanistic signaling data showing how this pathway functions after time-dependent blockade. I.e. It is hard to know what kind of cancers and which signaling pathways are most likely to be disrupted. Is there an effect on WASp Y291 protein phosphorylation? CDC42 phosphorylation? Cbl-P, T cell function? Can the authors show the drug has no effect on migration/ cell movement of CD45-non-hematopoietic cells such as breast cancer cell line that do not use WASp signalling access?

Minor Points

1. Title is a little ambitious: "Effective to Suppress" and "Therapeutic" are similar in concept; "novel" goes without saying in a Nat Comm paper. "Suppress" implies preventing outgrowth or preventing cancer whereas in vivo data shows a modest growth defect.
2. Fig S2 showing weak toxicity at 200uM. MNCs are large non-proliferative NK, mono and B cells. A better control would be CD34+
3. Best mechanistic data comes from flow cytometry LFA-1 activation. Can the raw flow plots show this data?
4. Insufficient detail for another researcher to perform the in silico screening. Please provide more detail on the parameters used and the input vs output algorithm for machine learning.
5. Fig 2B – nice work. Figure 2C FRET intensity of YTP distribution after SMC#13 treatment appears different. Can more examples be shown?
6. Figure 3 A,B no other flow cytometry or signalling data is shown with these primary samples.
7. Figure 7 does not tell us much about mechanism or best tumors to target WASp; Ki67 and Caspase 3 are relatively non-specific. May be best before or with Figure 6.
8. No statistics for Kaplan Meier survival, even if not significant, are given in results or on figure. Listed as $P < 0.001$ in legend. Any reason why NRG were used for survival and SCID/NODs for PK

and tumor volume? Any survival benefit in SCID/NOD transplanted mice?

9. Any effect on megakaryopoiesis or platelet activation? Any thrombocytopenia in the mice?

Reviewer #3 (Remarks to the Author):

Background

The authors have discovered a compound (SMC13) that enhances degradation of WASp, a key actin regulator, and have characterised this compound in vitro for target engagement, proliferation and motility inhibition and selectivity for activated haematopoietic cells and lack of toxicity, and in vivo for PK and therapeutic efficacy in haematopoietic cancer models.

Criticism and comments

1. Compound discovery:

One issue with the manuscript relates to hit/lead compound identification. On p7, the authors describe the use of the public structure of the N-WASP EVH1 domain (pdb: 2IFS) to run a virtual screen for compounds in silico. However the hit selected (SMC13) seems not to bind to N-WASP (Fig 1B, p8 l1-3), being selective for the WASp isoform vs N-WASP. There is further explanation required here why a hit designed by docking against N-WASP is inactive against its docking target but active for its homologue.

In addition, on p7 there are very little details provided on how the docking was done, what parameters were used for the software, what is the source of the library, how were the library and protein prepared for docking, what ranking functions were used. The authors should include this information. A figure of the putative docking of SMC13 and position in WIP peptide binding surface/pocket would be useful.

More details on the ML classifier - how it was trained, stats of model obtained should be described. It is also unusual that a hit resulted from a virtual screen was used as such in vivo, was any further chemistry carried out from hit to SMC13? Was the compound resynthesized for QC and validation? Has this compound been profiled in a panel of pharmacological targets for potential additional activities?

2. Mechanism of action:

The authors measure a Kd for the compound using MST (Fig 1A); an orthogonal biochemical assay is usually good practice to show convincing binding of inhibitor with target.

The authors show degradation of WASp in the presence of compound, and sensitivity of degradation to proteasome inhibitor MG132 (Fig2B and p9 first para). According to the hypothesis, if SMC13 inhibits interaction between WASp and WIP, there should be an increase in ubiquitination from negative control (second lane in blot, Fig 2B) to SMC13 treated (4th lane) in the presence of MG132 since the active Lys 76/81 are exposed in the presence on SMC13 and Ub is enhanced but the Ub-WASP is not degraded and should accumulate because the proteasome is inhibited. Fig 2B does not show that.

The authors show nice cellular data of inhibition of FRET signal between CFP-WASP and YFP-WIP, with complete lack of signal in the presence of inhibitor (Fig 2C). Has the potential of fluorescence quenching by SMC13 at 40uM been ruled out?

3. Toxicity assessment:

The authors show lack of target degradation in naïve PBMCs and degradation in activated PBMC and attribute this effect to partial unwinding of WIP-WASP interaction that allows compound access (p12, l259-264). This is an intriguing hypothesis and is supported by moderately enhanced degradation of WASp in activated PBMC, however no direct evidence is provided. It additionally raises the question of enhanced toxicity to PBMCs in both therapeutic and pathologic inflammation. In Figure S3B, the authors show that the compound does not induce cytokine release; however a more likely mechanism of toxicity based on WASp degradation in activated PBMCs is increased cytotoxic effect of SMC13 on cells stimulated with PHA (as in figure 3B positive control but also treated with SMC13). This has not

been determined or shown.

The authors also show limited toxicity in healthy (naïve) PBMC to doses up to 900 uM (p13-14 I299-305 and Fig S2). However the curve plateaus early at around 45%, has compound solubility been assessed since low solubility might confound the TD50 and would explain a curve as shown in Figure S2.

4. PK-PD correlation

The i.p. PK of SMC13 is assessed and Cmax of 97uM achieved at 100mg/kg (I341). However this peak concentration is fleeting, and at 1h, the concentration of SMC13 has dropped to ~15uM in plasma and 10uM in tumour, and decreasing further afterwards (Fig 5A). The degradation of WASp in cells (Jurkat) is shown after 12/24 hrs incubation at 40uM, it is not clear if this short exposure to compound in vivo can actually effectively induce degradation. What the paper is lacking is time-course of degradation of WASp with SMC13 in a panel of cell lines, WASp low/high or WASp dependent/independent including those used for the in vivo experiments.

Can the authors comment on the free drug concentration in blood vs same in cell experiments?

Intratumoural in vivo treatment is shown in figure S6. The compound concentration following IT administration should be measured to compare IP vs IT PK and level and timing of exposure. Although there is a hint of reduction of WASp in IT treatment, it is unclear if treatment with a SOC inhibitor would not have a similar non-specific effect.

The i.p. dosing is carried out at 72h difference, in 6 doses (Figure 5B-F); the remaining concentration of compound even at 24h, and even more so at 72h, will be close to zero. Unless sustained WASp degradation is shown following short exposure or the downstream effect is long-lived, it is unclear how proposed mechanism of action matches the efficacy seen. WASp reduction is shown in samples taken from tumours treated with SMC13 (Fig S6E, Fig 6A), how long after dosing is the sample taken – this should be clarified.

5. Target validation

A key set of data lacking from this paper is the orthogonal genetic validation of target with (inducible) siRNA of WASp or WIP, mutagenesis of the Lys76/81 or of residues critical for binding between WIP/WASP. Is any of the phenotypes shown in the Jurkat/Raji cells reproduced by knockdown? For example reduced migration, increased apoptosis, Ki-67 marker, reduced cell proliferation, dependence on activation of PBMC?

Summary

In summary, on the plus side, the mechanism of action proposed and exploited by the authors is very interesting, the data supporting engagement of this target by the compound discovered is reasonably convincing, there is interesting selectivity data for activated haematopoietic cells supporting an equally interesting mode of action, and the authors have characterised the compound in vivo and show interesting efficacy. The compound lack of some of the common toxicity mechanisms, although further clarification on other potential 'on-target' tox mechanisms is required. On the minus side, the compound discovery and profile (selectivity, physchem) is unclear, the genetic validation is lacking therefore efficacy via an alternative mechanism of the SMC13 compound cannot be ruled out and the PK-PD data is difficult to explain with current data.

Recommended Decision

Reject but further work might justify resubmission.

Point by point letter #NCOMMS-20-10509 entitled “Targeting WASp Provides a Therapeutic Approach for Hematopoietic Malignancies”.

REVIEWER#1

We thank the reviewer for his/her constructive comments.

"The authors use MST to determine the binding affinity of the small molecule to WASP. The error in the average fluorescence change is very high in these experiments....."

Thermophoretic effects are influenced by changes in size, charge, solvation energy, and conformation. The initial fluorescence is within the accepted variation standard. The use of whole cell lysates may induce small variations which appear large when graphed using this small thermophoretic scale. SMC #13, which showed a reasonable response amplitude, differs from the variation of the controls by a dose dependent response in the thermophoretic profile. In support of these findings, mutated WASp (YFP-WASp Δ WH1), did not bind. Furthermore, these data were verified with an orthogonal method using BLI, as presented in new Figure 1C, D.

We previously demonstrated that WASp undergoes conformational change upon cellular activation from an autoinhibited closed to an active open conformation (Fried et al Science Signaling, 2014). Here, we demonstrate that SMC#13 binds to the already opened conformation of WASp, not necessarily affecting the conformation; this may explain the relatively reasonable response amplitude.

"The authors claim that the small molecule specifically binds WASP, and imply that it interacts with the WH1 domain of WASP, but their data do not support this conclusion. Even if the MST data had less error, the results could reasonably be interpreted as the small molecule binding WASP indirectly through another protein because no experiments are done with purified WASP."

We added new data including BLI and MST with YFP-WASp Δ WH1 to support a direct binding mechanism (new Figure 1C-D). Furthermore, we now include new biochemical data showing that SMC#13 induces degradation of WT WASp but not of the mutated WASp Δ WH1, demonstrating the importance of the WH1 domain for targeted degradation (new Figure 2D)

"While the authors mention in the discussion that MST using lysate was the only way to measure the interaction because of problems purifying full length WASP, multiple groups have been able to purify and study the WH1 domain in isolation. Given that small molecule is hypothesized to bind to WH1, the authors should be able to use another method (e.g, ITC) to more convincingly show the interaction (and measure the affinity) between the small molecule and the purified WH1 domain."

As recommended by the reviewer, we produced a YFP-WH1 construct for MST analysis; however, as shown below, we observed spontaneous cleavage between the WH1 fragment and the YFP tag. Therefore, as an alternative approach, we conducted BLI and MST with a WASp mutant lacking the WH1 domain (new Figure 1B and D). Furthermore, we performed stability experiments with the wt WASp vs. the Δ WH1 mutant, demonstrating that SMC#13 only induces the degradation of the full-length WT WASp and not the Δ WH1 mutant (Figure 2D).

"In addition, more information about the docking and the small molecule identification should be included in figure 1. For instance, the authors should show the theoretical structure of the docked small molecule bound to the WH1 domain. Based on this model,

the authors should comment on why the small molecule doesn't bind to the WH1 of N-WASP in the discussion. In the results section the authors should mention that WAVE doesn't have a WH1 domain, so it is not surprising that the small molecule wouldn't bind to WAVE based on their mechanism (Figure 1C). The authors should also show the chemical structure of the small molecule in figure 1."

As suggested by the reviewer, a scheme detailing the *in-silico* screening process (new Figure S1A) and a putative binding model (new Figure S1B-D) were added to the manuscript. Furthermore, details regarding the virtual screening were added to the text (Results section p.7-8). We emphasized the use of virtual mutagenesis on the known WH1 domain model of N-WASp (PDB ID: 2IFS) to screen for SMCs that specifically bind to WASp (page 7, lines 142-144). A sentence describing lack of binding to WAVE2 was added to the Results section (page 9, lines 189-191). The chemical structure of SMC#13 is now shown in Figure S1E.

"A key conclusion of the study is that the small molecule stimulates degradation of WASP by competing with WIP for WASP binding. To support this conclusion, the authors should use competition binding assays with purified components, as described above."

Since WASp association to WIP requires the N' and C' termini of WASp (Fried et al, 2014), we cannot precipitate WIP with only the WH1 domain of WASp. In addition to the FRET data that clearly shows loss of WASp binding to WIP following SMC#13 treatment, we show through the BLI and MST experiments that the WASP Δ WH1 mutant does not bind SMC#13. This is further corroborated with biochemical experiments that demonstrate that the cellular concentration of WASp Δ WH1 is unaffected by SMC#13 in contrast to that of WT WASp. Collectively these experiments show that the WH1 domain of WASp, which is masked and protected by WIP, is the target of SMC#13, which interferes with WIP:WASp binding and promotes WASp degradation.

"The description of the influence of the small molecule inhibitor on WASP is confusing because of the use of the phrase "WASP expression". The authors' use of this phrase instead of "the cellular concentration of WASP" seems in some places to be inconsistent with their results. For instance, on line 170 they write "... WASp expression was almost completely abrogated." To most readers, this will probably be interpreted as meaning that either the transcription or translation of WASP was decreased, whereas the authors' data suggest that it is instead that the small molecule stimulates degradation of WASP to decrease the its steady state cellular concentration."

Thank you for this comment; we now write "WASp cellular concentration".

"The authors should address why the concentration of the small molecule required to influence WASP concentrations in cells is many orders of magnitude higher than the affinity of the molecule for WASP. In addition, the author should discuss whether or not the small molecule has properties consistent with a drug (e.g. logP, mw, number of h-bond donors and acceptors). In addition, while it is stated at least once in the text, each figure legend should mention the concentration of the small molecule used in the experiments."

The possible explanations for the difference between the in-vitro concentrations and the Kd value are now discussed in the text (please see page 21, lines 479-484). This discrepancy might be a result of moderate solubility and high lipophilicity of SMC #13, which could reduce its bioavailability in the cellular context in comparison to small volumes of cellular lysates. We also speculate that the strong interaction between WIP and WASp under physiological conditions requires higher concentrations of SMC #13. Another possible

explanation might be the effects of fluorescent tags in the N' terminus of WASp in the binding assays; these may weaken the interaction between WIP and the N' terminal WH1 domain of WASp. Furthermore, as requested we added table #1 to summarize the properties of the SMC#13.

"The sentence on line 259 is unclear. What do the authors mean by ‘‘partial dissociation’’ of WIP and WASP?"

We previously demonstrated that C' terminus of WASp associates with the N' terminus of WIP, and that the N' terminus of WASp associates with the C' terminus of WIP. Upon cellular activation, a partial disassociation occurs between the N' terminus of WASp and the C' terminus WIP, leaving the other binding site intact (Fried et al. Science Signaling, 2014). This issue is now clarified in the text (please see Introduction section).

"For the data in figure 4A, the authors should report a non-normalized concentration of WASP, since comparing the concentrations between stimulated and unstimulated cells would be useful. Related to this, the data in figure 4D seem difficult to rationalize. Why is there little to no motility in the stimulated cells treated with the compound when the concentration of WASP in these cells looks very similar to the concentration of WASP in the unstimulated cells?"

As requested by the reviewer, new Figure 4A showing non-normalized cellular concentration of WASp between the activated and the non-activated cells is now presented. We observed an upregulation of WASp cellular concentration (increase of approximately 44%) after continuous stimulation with PMA and Ionomycin. Incubation of the stimulated PBMCs with SMC #13 decreases the cellular concentration of WASp (a reduction of approximately 28% in unstimulated cells, and a reduction of ~72% in stimulated cells) which impairs directional movement of the cells upon chemokine stimulation (as shown by Jain et al. 2015).

**"Much of the discussion is redundant, in my view. Specifically:
- the first paragraph in the discussion re-establishes the need to develop better treatments for hematopoietic malignancies. This is established in the introduction so in is unnecessary in the discussion.
- Much of the discussion is a recap of the results. The authors could make the discussion more succinct by not recapping the results except to the extent necessary to discuss implications of the results that were not mentioned in the results."**

As suggested, the Discussion was significantly revised.

"The text in the discussion describing the rationale for using MST in lysates is somewhat misleading, since, as pointed out above, experiments could be done with the isolated WH1 domain."

Please see our response regarding the WH1 fragment above. We added BLI, a method that is orthogonal to MST, and various biochemical experiments to strengthen the data (new Figures 1 and 2D).

"Lines 469-473 are also somewhat misleading, in my view. The statement that ‘‘malignant cells. mainly express the open, activated form of WASp while naïve cells dominantly express the closed-autoinhibited form’’ is referenced with a paper that is purely biochemical/in vitro and doesn't compare malignant versus naïve cells. The authors should clearly state here that this statement is their speculation. Further, in my view, it is inaccurate to say that cells ‘‘express’’ one form or the other. The autoinhibited versus activated states are in equilibrium with one another. It's possible that the open

conformation is more populated in the malignant cells, but it doesn't make sense to me to say that the malignant cells "express the open, activated form"."

This point is well taken, and the text was revised accordingly (please see pages 14 and 21). Our model, while based on experimental data and the literature is now described as speculative in the Discussion.

"Related to the above comment, the authors seem to be using the concept of WASP open and closing to indicate 1.) exposure of ubiquitylation sites on the WH1 domain and 2.) release of the VCA segment of WASP from the GBD of WASP. What is the evidence that these two molecular events occur in concert?"

The molecular events surrounding WASp activation and degradation have been extensively studied (Reicher et al. 2012, Fried et al. 2014). Upon cellular stimulation, WASp is recruited to the cellular membrane, and its VCA domain is exposed in order to recruit Arp2/3. Reicher et al. 2012 showed that upon cellular stimulation, the TKB domain of the Cbl proteins associates with tyrosine 291 of WASp to induce ubiquitylation of lysine 76 and 81 of WASp, leading to its degradation. These results were also corroborated by Watanabe et al. 2013, who showed accumulation of cleaved WASp within 15 minutes following TCR activation.

WASP degradation is dependent on the partial disassociation of WASp from the C' terminal of WIP, leading to the exposure of the ubiquitylation site (Fried et al 2014, Reicher et al. 2012). Furthermore, we demonstrated that WIP phosphorylation at serine 488 promotes WASp-WIP disassociation (Fried et al., 2014).

"In figure 8, why are the WH2 domains of WIP depicted as binding to the VCA segment of WASP?"

As mentioned above, WIP and WASp bind in an antiparallel orientation (Fried et al. 2014.). This is now further clarified in the Discussion. Please see pages 21-22, lines 492-497.

REVIEWER #2.

We thank the reviewer for his/her comments, and we believe that addressing them has significantly improved our manuscript.

"1] Thus evidence of further on target specific could be improved by at least one of the following:-

Four of the seven options suggested by the reviewer were accomplished to further support target binding.

"(i) further detail and modelling of the *in-silico* screen approach showing docking"

More details describing the *in-silico* screening process, including the machine learning trained classifier, are now provided in the manuscript (please see pages 7-8). Furthermore, new Figure S1 was also added to provide more details.

"(ii) performance of MST with mutated WASp lacking residues adjacent to lysine 76 and 81 that are predicted to interact with the molecule"

MST and BLI analysis using YFP-WASP wt vs. Δ WH1 are now included (new Fig. 1A-D).

"(iii) performance of an orthogonal binding method such as BLI, SPR, ITC"

As mentioned in the comment above, we incorporated BLI data (Octet, FortéBio) in addition to the MST (new Figure 1C-D).

"(iv) demonstration that covalently tagged SMC#13 can pull down WASp"

As establishing this system is highly time consuming, other alternatives were chosen.

"(v) demonstration that a modified version of SMC#13 (enantiomer or removal of benzyl-hydroxyl) does not bind WASp or result in WASp degradation"

Since SMC modifications are time consuming and require structural determination, we adapted the other suggested options.

"(vi) demonstration that other actin-depolymerizing drugs do not change WASp protein degradation stability."

These experiments were performed, and the data were added to the manuscript (new Figure S1D). We assessed WASp expression in T-ALL cells treated with either SMC#13 or cytochalasin D. While SMC#13 significantly decreased WASp protein levels, cytochalasin D had no observable effect.

"(vii) demonstration that GFP- or CYP-WASp bind with the same Kd"

To address this issue, we chose to validate nonspecific binding to the fluorophore by adding a YFP-only control sample to the MST and BLI experiments. This strategy was performed in order to avoid exchanging the fluorophore, and to maintain the same tag in all other control samples. Note that there was no detectable binding of SMC#13 to the fluorophore (new Figure S2C).

"2] Which NHL sub-type do the authors predict will be most sensitive? Follicular or Burkitt? T-ALL vs B-ALL? What about M7 AML?"

The exact malignant subtypes of the cells that were used are now detailed. Since WASp plays a critical role in all hematopoietic cell lineages, we assume that the effects induced by SMC#13 on lymphoproliferative cell lines and primary cells in this study may be relevant to a very broad range of hematological malignancies. This approach may be applicable not only to lymphocytic malignancies, but to myeloid malignancies, as well. This should be examined in the future. This point is now clarified in the text (please see pages 5-6 and 23).

"3] Is there an effect on WASp Y291 protein phosphorylation? CDC42 phosphorylation? Cbl-P, T cell function? Can the authors show the drug has no effect on migration/ cell movement of CD45- non-hematopoietic cells such as breast cancer cell line that do not use WASp signaling access?"

In general, as correctly noted by the reviewer, the role of WASp and its signaling pathway in hematological malignancies is poorly understood and requires extensive study. Since the revised manuscript is already very lengthy (7 main and 9 supplementary figures) we believe that the possible effect of SMC#13 on signal transduction mechanisms is out of the scope of this paper.

However, we did exclude, the effect of SMC #13 on non-hematopoietic cells by assessing the migration of melanoma cell line (MEL1106) (new Figure S4).

"Minor Points"

1. Title is a little ambitious: "Effective to Suppress" and "Therapeutic" are similar in concept; "novel" goes without saying in a Nat Comm paper. "Suppress" implies preventing outgrowth or preventing cancer whereas in vivo data shows a modest growth defect."

The title was changed based on the reviewer's comment.

"2. Fig S2 showing weak toxicity at 200uM. MNCs are large non-proliferative NK, mono and B cells. A better control would be CD34+"

Although we used XTT viability to quantify cellular growth and measure toxicity, we believe that evaluating the toxic effect of this compound does not require use of highly proliferative cells; thus we further assessed cell viability by PI staining (new Figure S5).

"3. Best mechanistic data comes from flow cytometry LFA-1 activation. Can the raw flow plots show this data?"

The raw flow plots are now shown in new Figure 2G

"4. Insufficient detail for another researcher to perform the *in-silico* screening. Please provide more detail on the parameters used and the input vs output algorithm for machine learning."

More details describing the *in-silico* screening process, including the docking and the machine learning trained classifier are now included, and a new section entitled "*in-silico* screening for WASp-binding SMCs" was added to the Results and Materials & Methods sections (please see page 7-8, 24-25 and new Figure S1).

"5. Fig 2B – nice work. Figure 2C FRET intensity of YTP distribution after SMC#13 treatment appears different. Can more examples be shown?"

Another representative image was added to replace the older one (new Figure 2C).

"6. Figure 3 A,B no other flow cytometry or signaling data is shown with these primary samples."

Please see our response above, to reviewer #2, point #3.

"7. Figure 7 does not tell us much about mechanism or best tumors to target WASp; Ki67 and Caspase 3 are relatively non-specific. May be best before or with Figure 6."

The IHC results that were shown in old Figure 7 are now merged with Figure 6.

"8. No statistics for Kaplan Meier survival, even if not significant, are given in results or on figure. Listed as $P < 0.001$ in legend. Any reason why NRG were used for survival and SCID/NODs for PK and tumor volume? Any survival benefit in SCID/NOD transplanted mice?"

The statistics using log rank test were calculated, and are now added to the figure.

Both mice strains are immune incompetent. All the mice that were used for in-vivo models were Nonobese diabetic/severe combined immunodeficiency mice, which are commonly used as models for human hematological malignancy. No differences were observed between the two strains. Each set of experiments was consistently conducted in the same strain.

"9. Any effect on megakaryopoiesis or platelet activation? Any thrombocytopenia in the mice?"

These were not determined. In this work we mainly focused on lymphoproliferative diseases.

Reviewer #3

We thank the reviewer for his/her comments and suggestions.

Criticism and comments

"1. Compound discovery:

One issue with the manuscript relates to hit/lead compound identification. On p7, the

authors describe the use of the public structure of the N-WASP EVH1 domain (pdb: 2IFS) to run a virtual screen for compounds in silico. However the hit selected (SMC13) seems not to bind to N-WASP (Fig 1B, p8 11-3), being selective for the WASp isoform vs N-WASP. There is further explanation required here why a hit designed by docking against N-WASP is inactive against its docking target but active for its homologue.

To this day, no structural model of WASp, or even its WH1 domain, was published. Thus, we utilized the published 3D NMR structure of N-WASP as the basis of the virtual screening. We now describe details regarding the preparation of the WASp model from the 2IFS PDB model (page 7, lines 142-144) and explain the use of the virtual mutagenesis tool in the Discovery Studio, to modify the 2IFS model using the sequence of WASp. The virtual docking and subsequent machine learning-based prediction were performed using the modified model as a surrogate for the WASp WH1 domain. This enabled the selection of a small molecular compound based on its binding to WASp. Lack of off-target binding was experimentally verified utilizing MST and BLI analyses. Moreover, no effect of SMC13 was seen using migration assay of melanoma cells, which exclusively express N-WASP, and **not** WASp.

In addition, on p7 there are very little details provided on how the docking was done, what parameters were used for the software, what is the source of the library, how were the library and protein prepared for docking, what ranking functions were used. The authors should include this information. A figure of the putative docking of SMC13 and position in WIP peptide binding surface/pocket would be useful. More details on the ML classifier - how it was trained, stats of model obtained should be described."

As requested by the reviewer, the required information was added and is now integrated into the manuscript (page 7-8). The text now details the procurement of the virtual libraries and the virtual docking process, and explains the training and the use of the machine learning based classifier. Since the classifier forms the topic of another manuscript (Feiglin A et al. In preparation), we are precluded from publishing it and its core statistics (e.g. a precision-recall curve) in this manuscript.

A putative figure of the WASp-SMC#13 structure was added to the manuscript (Figure S1A-D).

"It is also unusual that a hit resulted from a virtual screen was used as such in vivo, was any further chemistry carried out from hit to SMC13? Was the compound resynthesized for QC and validation? Has this compound been profiled in a panel of pharmacological targets for potential additional activities?"

We profiled some pharmacological aspects of the compound using the SwissADME web tool (Table 1 and Figure S1F) and validated the specific binding to WASp and not to other closely-related homologs. No further chemical analysis was performed. However, it is important to note that SMC13 was one of the SMCs that were discovered as potential modulators of WASp. The core goal of this manuscript was to demonstrate the concept that induction of WASp degradation may suppress hematological malignancies. Obviously, for drug development and clinical applications, optimization of SMC13 is warranted.

2. Mechanism of action:

The authors measure a Kd for the compound using MST (Fig 1A); an orthogonal biochemical assay is usually good practice to show convincing binding of inhibitor with target."

Our data were further validated with an orthogonal method using BLI, which is now included in the manuscript (Figure 1C and D). We observed SMC13 binding to wt WASp but not to the Δ WH1 mutant.

"The authors show degradation of WASp in the presence of compound, and sensitivity of degradation to proteasome inhibitor MG132 (Fig2B and p9 first para). According to the hypothesis, if SMC13 inhibits interaction between WASp and WIP, there should be an increase in ubiquitination from negative control (second lane in blot, Fig 2B) to SMC13 treated (4th lane) in the presence of MG132 since the active Lys 76/81 are exposed in the presence on SMC13 and Ub is enhanced but the Ub-WASp is not degraded and should accumulate because the proteasome is inhibited. Fig 2B does not show that."

As can be seen in the control lanes (lane 1 and 2) there is residual ubiquitylation in the first lane, and the ubiquitylation increases in the second lane with MG132 treatment. When we added SMC #13 we observed degradation of WASp and of its ubiquitylated form (third lane). Treatment of the cells with SMC #13 and MG132 restored WASp expression, but WASp did not accumulate to the same level as that of the negative control cells treated with MG132 alone. It is important to note that since degradation is visualized as a continuous smear and not as a clear band, quantification of ubiquitination is very difficult to conduct by densitometry. In addition, an alternative pathway of WASp degradation occurs through Calpain, which is not proteasome dependent, but it is influenced by the WASp-WIP interaction, as demonstrated in the work of de la Fuente et al. 2006 and chou et al. 2006. Since SMC#13 affected the WIP;WASp equilibrium, it is possible that it also promotes calpain-dependent cleavage of WASp; however, this is yet to be determined.

"The authors show nice cellular data of inhibition of FRET signal between CFP-WASp and YFP-WIP, with complete lack of signal in the presence of inhibitor (Fig 2C). Has the potential of fluorescence quenching by SMC13 at 40uM been ruled out?"

No indication of fluorescence quenching by SMC13 was detected. As observed by the molecular imaging experiments, no significant differences were measured between the fluorescent intensities of the controls and samples containing SMC #13 at 40µM. The only differences that were observed were in the donor-sensitized acceptor emission channel. Furthermore, in the MST and BLI experiments, there were no differences in the fluorescence intensity signal of the YFP-only sample (YFP is the acceptor of the FRET experiments) when treated with vehicle or SMC13.

"3. Toxicity assessment:

The authors show lack of target degradation in naïve PBMCs and degradation in activated PBMC and attribute this effect to partial unwinding of WIP-WASp interaction that allows compound access (p12, 1259-264). This is an intriguing hypothesis and is supported by moderately enhanced degradation of WASp in activated PBMC, however no direct evidence is provided. It additionally raises the question of enhanced toxicity to PBMCs in both therapeutic and pathologic inflammation. In Figure S3B, the authors show that the compound does not induce cytokine release; however a more likely mechanism of toxicity based on WASp degradation in activated PBMCs is increased cytotoxic effect of SMC13 on cells stimulated with PHA (as in figure 3B positive control but also treated with SMC13). This has not been determined or shown."

Based on our experience in studying lymphocyte activation, we believe that there is a fundamental difference in lymphocyte activation during infection vs. the constitutive activation in lymphoproliferative diseases. We postulate that SMC#13 more strongly impacts cancerous cells as compared to healthy activated cells due to the time window of its activity, i.e. the period during which WASp is open and unprotected by the WIP chaperon is significantly larger in transformed cells. Obviously, in order to advance SMC#13 toward

clinical testing as a therapeutic agent for hematological malignancies, these issues should be further investigated. Moreover, based on our data, SMC#13 does not aggravate cytokine secretion.

"The authors also show limited toxicity in healthy (naïve) PBMC to doses up to 900 uM (p13-14 1299-305 and Fig S2). However the curve plateaus early at around 45%, has compound solubility been assessed since low solubility might confound the TD50 and would explain a curve as shown in Figure S2."

Another method to evaluate cytotoxicity using PI staining is now included in the manuscript (Figure S5).

"4. PK-PD correlation

The i.p. PK of SMC13 is assessed and Cmax of 97uM achieved at 100mg/kg (1341). However this peak concentration is fleeting, and at 1h, the concentration of SMC13 has dropped to ~15uM in plasma and 10uM in tumour, and decreasing further afterwards (Fig 5A). The degradation of WASp in cells (Jurkat) is shown after 12/24 hrs incubation at 40uM, it is not clear if this short exposure to compound in vivo can actually effectively induce degradation. What the paper is lacking is time-course of degradation of WASp with SMC13 in a panel of cell lines, WASp low/high or WASp dependent/independent including those used for the in vivo experiments."

Since we achieved effective concentration of 97.2 μ M in the tumors after 30 min, which is more than double the effective 40 μ M concentration tested *in-vitro*, we believe that this concentration is more than sufficient to affect WASp cellular levels. Moreover, the concentration in tumors after 30 min is below the TD50 value. The *ex-vivo* data presented in Figure 6A demonstrate that WASp is degraded in the tumor cells 24h after systemic injection of SMC#13. Time-course evaluation of WASp stability within short time periods presented in Figure 2E indicates that WASp already starts to degrade after a short 2h incubation. These evidences indicate that the decrease of WASp cellular concentration in the tumor cells for such a time period might downregulate WASp-dependent cellular functions for at least 24h attenuating *in-vivo* tumor growth. Further testing of short-term treatment regimen might be more effective for tumor attenuation and prolong mice survival.

Our data indicate that upregulation of WASp ubiquitylation and degradation in cancer cells might eventually abolish key cellular functions, thereby attenuating *in vivo* tumor growth.

"Can the authors comment on the free drug concentration in blood vs same in cell experiments?"

Following injection of a 100mg/kg dose, the SMC concentration in the plasma was 116 \pm 20 μ M which is within the effective range according to our *in-vitro* experiments.

However, as mentioned above, it is also below the estimated TD50 value. based on all these data, we anticipate that this dosage should be sufficient according to the PK and efficacy data, but we do not rule out the possibility that shortening the intervals between the injections might improve efficacy and survival.

"Intratumoural in vivo treatment is shown in figure S6. The compound concentration following IT administration should be measured to compare IP vs IT PK and level and timing of exposure....."

The IT experiments were conducted as a proof of concept for the ability of SMC#13 to mitigate tumor growth prior to IP dosage calibration. Obviously, since the IP injections are the preferred and more common administration route, all *ex-vivo* experiments were based on this system.

"The i.p. dosing is carried out at 72h difference, in 6 doses (Figure 5B-F); the remaining concentration of compound even at 24h, and even more so at 72h, will be close to zero. Unless sustained WASp degradation is shown following short exposure or the downstream effect is long-lived, it is unclear how proposed mechanism of action matches the efficacy seen. WASp reduction is shown in samples taken from tumours treated with SMC13 (Fig S6E, Fig 6A), how long after dosing is the sample taken – this should be clarified."

Please see above, point#4, our answer to this question. In addition, the details regarding the time of tumor extraction are now mentioned on page 18, lines 415-416. Furthermore, we anticipate that an additive long term functional effect occurs after sequential exposures of the cancerous cells to SMC#13.

"5. Target validation

A key set of data lacking from this paper is the orthogonal genetic validation of target with (inducible) siRNA of WASp or WIP, mutagenesis of the Lys76/81 or of residues critical for binding between WIP/WASP. Is any of the phenotypes shown in the Jurkat/Raji cells reproduced by knockdown? For example reduced migration, increased apoptosis, Ki-67 marker, reduced cell proliferation, dependence on activation of PBMC?"

We now include orthogonal validation using the WASp Δ WH1 mutant in MST, BLI and biochemical analysis (Figures 1 and 2D). These results show that WASp lacking the WH1 domain does not bind SMC#13 and is resistant to degradation.

We currently have data regarding the effects of WASp silencing in hematological malignancies; however, we anticipate publishing these results as part of a separate manuscript. Indeed, as noted by the reviewer, knockdown of WASp using siRNAs results in similar effect as SMC#13. Indeed, Murga-Zamalloa et al, 2016 showed that WASp expression correlates with severity of anaplastic large cell lymphoma. Similar results are shown in Figure S9.

REVIEWER COMMENTS

Reviewer #1 (Remarks to the Author):

I am satisfied with the authors' response to the reviews and their revised manuscript.

Reviewer #2 (Remarks to the Author):

All questions have been adequately addressed in the revised manuscript.

There is still a discrepancy between the concentrations used in cell assays and Western blot vs binding affinity in Fig 1A vs binding affinity in BLI experiments but the authors give some explanation for this in the discussion.

Reviewer #3 (Remarks to the Author):

Please see below this reviewers' comments IN CAPITALS to differentiate the text from the original review and the authors rebuttal and amended manuscript

SUMMARY OF KEY COMMENTS

THE MANUSCRIPT IS IMPROVED BY THE ADDITION OF ORTHOGONAL ASSAY DATA AND WH1 DELETION MUTANT, AS WELL AS CELLULAR DATA IN MELANOMA CELLS EXPRESSING N-WASP TO VALIDATE SPECIFIC BINDING OF SMC13 TO WASP. THE DOCKING PROCESS IS CLARIFIED.

THERE ARE HOWEVER ASPECTS OF THE PAPER THAT STILL REMAIN TO BE ADDRESSED, IN PARTICULAR: DOCKING OF A BELIEVABLE CONFORMATION OF SMC13, AND COMPOUND CHARACTERIZATION FOR IDENTITY AND PURITY. THESE MUST BE ADDRESSED BEFORE THE PAPER IS CONSIDERED FOR PUBLICATION.

IN ADDITION, A NUMBER OF SIMPLE EXPERIMENTS WILL HELP IMPROVE THE UNDERSTANDING OF COMPOUND BEHAVIOR IN IN VITRO AND IN VIVO ASSAYS: SOLUBILITY AND PLASMA PROTEIN BINDING OF THE COMPOUND, AND ASSESSING THE EFFECT OF SHORT TERM INCUBATION OF COMPOUND ON CELLULAR WASP DEGRADATION. FURTHER PRELIMINARY GENETIC VALIDATION DATA WOULD ALSO SUPPORT THE ON-TARGET MECHANISM PROPOSED.

THIS REVIEWER HAS ADDED INDIVIDUAL COMMENTS IN THE RELEVANT SECTIONS BELOW:

Reviewer #3

We thank the reviewer for his/her comments and suggestions.

Criticism and comments

"1. Compound discovery:

One issue with the manuscript relates to hit/lead compound identification. On p7, the authors describe the use of the public structure of the N-WASP EVH1 domain (pdb: 2IFS) to run a virtual screen for compounds in silico. However the hit selected (SMC13) seems not to bind to N-WASp (Fig 1B, p8 l1-3), being selective for the WASp isoform vs N-WASp. There is further explanation required here why a hit designed by docking against N-WASp is inactive against its docking target but active for its homologue.

To this day, no structural model of WASp, or even its WH1 domain, was published. Thus, we utilized the published 3D NMR structure of N-WASp as the basis of the virtual screening. We now describe details regarding the preparation of the WASp model from the 2IFS PDB model (page 7, lines 142-144) and explain the use of the virtual mutagenesis tool in the Discovery Studio, to modify the 2IFS model using the sequence of WASp.

THIS IS INTERESTING AND EXPLAINS THE RESULTS; A BRIEF LISTING OF N-WASP

RESIDUES REPLACED WITH CORRESPONDING WASP RESIDUES USING THE BIOVIA MUTAGENESIS TOOL WOULD BE USEFUL.

The virtual docking and subsequent machine learning-based prediction were performed using the modified model as a surrogate for the WASp WH1 domain. This enabled the selection of a small molecular compound based on its binding to WASp. Lack of off-target binding was experimentally verified utilizing MST and BLI analyses. Moreover, no effect of SMC13 was seen using migration assay of melanoma cells, which exclusively express N-WASp, and not WASp.

In addition, on p7 there are very little details provided on how the docking was done, what parameters were used for the software, what is the source of the library, how were the library and protein prepared for docking, what ranking functions were used. The authors should include this information. A figure of the putative docking of SMC13 and position in WIP peptide binding surface/pocket would be useful.

More details on the ML classifier - how it was trained, stats of model obtained should be described." As requested by the reviewer, the required information was added and is now integrated into the manuscript (page 7-8). The text now details the procurement of the virtual libraries and the virtual docking process, and explains the training and the use of the machine learning based classifier. Since the classifier forms the topic of another manuscript (Feiglin A et al. In preparation), we are precluded from publishing it and its core statistics (e.g. a precision-recall curve) in this manuscript.

THE QUESTION REGARDING DOCKING WORKFLOW IS ADDRESSED BY THE ADDITIONAL INFORMATION PROVIDED. CLEARLY THE PUBLICATION OF THE ADDITIONAL MANUSCRIPT WILL BE REQUIRED FOR THE WORK TO BE REPRODUCIBLE BY OTHER COMPUTATIONAL SCIENTISTS.

A putative figure of the WASp-SMC#13 structure was added to the manuscript (Figure S1A-D).

THERE IS A PROBLEM WITH THE 2,3-DIHYDRO-1H-INDEN-2-YL (INDANE) STRUCTURE SHOWN ON DOCKING IN FIG S1B, S1C: THE ANGLE BETWEEN THE 5-MEMBER RING AND THE PHENYL FUSED TO IT APPEARS TO BE AROUND 90 DEGREE IN THE PICTURE OF DOCKED COMPOUND SMC13. BASED ON THE STRUCTURE SHOWN IN FIGURE S1E, THE BICYCLIC INDANE STRUCTURE IS IN REALITY ALMOST PLANAR BECAUSE OF THE PLANARITY OF THE PHENYL RING, AND IN PARTICULAR THE TWO C-C BONDS LEADING FROM THE PHENYL RING OF THE INDANE SHOULD BE IN THE SAME PLANE. THIS 3D CONFORMATION OF THE DOCKED STRUCTURE OF SMC13 THEREFORE APPEARS WRONG AND IS NOT BELIEVABLE CHEMICALLY, RAISING QUESTIONS REGARDING THE PROPOSED INTERACTIONS IN THE BINDING LOCATION OF COMPOUND SMC13.

FURTHERMORE, THE AUTHORS COMMENT ON THE BINDING OF THE INHIBITOR (LINES 479-484) AS SUPPORTING THEIR MECHANISTIC HYPOTHESIS. GIVEN THE QUESTION RAISED BY THE DOCKED STRUCTURE ABOVE AND THE FACT THAT THIS IS A DOCKING MODEL, NOT AN ACTUAL EXPERIMENTALLY DERIVED DATA (E.G X-RAY CRYSTAL OR NMR STRUCTURE), THESE COMMENTS DO NOT STAND.

"It is also unusual that a hit resulted from a virtual screen was used as such in vivo, was any further chemistry carried out from hit to SMC13? Was the compound resynthesized for QC and validation? Has this compound been profiled in a panel of pharmacological targets for potential additional activities?"

We profiled some pharmacological aspects of the compound using the SwissADME web tool (Table 1 and Figure S1F) and validated the specific binding to WASp and not to other closely-related homologs. No further chemical analysis was performed. However, it is important to note that SMC13 was one of the SMCs that were discovered as potential modulators of WASp. The core goal of this manuscript was to demonstrate the concept that induction of WASp degradation may suppress hematological malignancies. Obviously, for drug development and clinical applications, optimization of SMC13 is warranted.

ACCORDING TO THE JOURNAL INSTRUCTIONS

([HTTPS://WWW.NATURE.COM/NCOMMS/JOURNAL-POLICIES/EDITORIAL-PUBLISHING-POLICIES](https://www.nature.com/ncomms/journal-policies/editorial-publishing-policies)), 'AUTHORS MUST PROVIDE ADEQUATE DATA TO SUPPORT THEIR ASSIGNMENT OF IDENTITY AND PURITY FOR EACH NEW COMPOUND DESCRIBED IN THE MANUSCRIPT. AUTHORS SHOULD PROVIDE A STATEMENT CONFIRMING THE SOURCE, IDENTITY AND

PURITY OF KNOWN COMPOUNDS THAT ARE CENTRAL TO THE SCIENTIFIC STUDY, EVEN IF THEY ARE PURCHASED OR RESYNTHESIZED USING PUBLISHED METHODS.' SINCE THE COMPOUND SMC13 IS KEY TO THE RESEARCH AND THE AUTHORS HAD ACCESS TO REASONABLY LARGE AMOUNTS OF COMPOUND (THE IN VIVO THERAPY ALONE REQUIRED ABOUT 200 MG OF COMPOUND), IS THERE A CERTIFICATE OF ANALYSIS/H-NMR/LCMS CHARACTERISATION OF THE COMPOUND TO CONFIRM THAT THE PROPOSED STRUCTURE USED IN BIOCHEMICAL EXPERIMENTS IS CORRECT AND OF SUFFICIENT PURITY?

2. Mechanism of action:

The authors measure a Kd for the compound using MST (Fig 1A); an orthogonal biochemical assay is usually good practice to show convincing binding of inhibitor with target."

Our data were further validated with an orthogonal method using BLI, which is now included in the manuscript (Figure 1C and D). We observed SMC13 binding to wt WASp but not to the Δ WH1 mutant. THE NEW DATA STRENGTHEN THE EVIDENCE OF COMPOUND INTERACTION WITH WT WASP AND DEMONSTRATE DOSE-DEPENDENT BINDING. THE AUTHORS SHOULD REPORT THE KD ESTIMATED BY BLI, FOR COMPARISON TO MST KD.

"The authors show degradation of WASp in the presence of compound, and sensitivity of degradation to proteasome inhibitor MG132 (Fig2B and p9 first para). According to the hypothesis, if SMC13 inhibits interaction between WASp and WIP, there should be an increase in ubiquitination from negative control (second lane in blot, Fig 2B) to SMC13 treated (4th lane) in the presence of MG132 since the active Lys 76/81 are exposed in the presence on SMC13 and Ub is enhanced but the Ub-WASp is not degraded and should accumulate because the proteasome is inhibited. Fig 2B does not show that."

As can be seen in the control lanes (lane 1 and 2) there is residual ubiquitylation in the first lane, and the ubiquitylation increases in the second lane with MG132 treatment. When we added SMC #13 we observed degradation of WASp and of its ubiquitylated form (third lane). Treatment of the cells with SMC #13 and MG132 restored WASp expression, but WASp did not accumulate to the same level as that of the negative control cells treated with MG132 alone. It is important to note that since degradation is visualized as a continuous smear and not as a clear band, quantification of ubiquitination is very difficult to conduct by densitometry. In addition, an alternative pathway of WASp degradation occurs through Calpain, which is not proteasome dependent, but it is influenced by the WASp-WIP interaction, as demonstrated in the work of de la Fuente et al. 2006 and chou et al. 2006. Since SMC#13 affected the WIP;WASp equilibrium, it is possible that it also promotes calpain-dependent cleavage of WASp; however, this is yet to be determined.

THIS REVIEWER AGREES THAT THE COMPOUND DOES ACHIEVE THE PURPOSE OF DEGRADING WASP, EVEN IF THE EXACT MECHANISM IS NOT FULLY ELUCIDATED.

"The authors show nice cellular data of inhibition of FRET signal between CFP-WASp and YFP-WIP, with complete lack of signal in the presence of inhibitor (Fig 2C). Has the potential of fluorescence quenching by SMC13 at 40uM been ruled out?"

No indication of fluorescence quenching by SMC13 was detected. As observed by the molecular imaging experiments, no significant differences were measured between the fluorescent intensities of the controls and samples containing SMC #13 at 40 μ M. The only differences that were observed were in the donor-sensitized acceptor emission channel. Furthermore, in the MST and BLI experiments, there were no differences in the fluorescence intensity signal of the YFP-only sample (YFP is the acceptor of the FRET experiments) when treated with vehicle or SMC13.

THIS REVIEWER AGREES THAT FLUORESCENCE QUENCHING DOES NOT APPEAR TO BE AN ISSUE.

"3. Toxicity assessment:

The authors show lack of target degradation in naïve PBMCs and degradation in activated PBMC and attribute this effect to partial unwinding of WIP-WASp interaction that allows compound access (p12, l259-264). This is an intriguing hypothesis and is supported by moderately enhanced degradation of WASp in activated PBMC, however no direct evidence is provided. It additionally raises the question of enhanced toxicity to PBMCs in both therapeutic and pathologic inflammation. In Figure S3B, the authors show that the compound does not induce cytokine release; however a more likely mechanism

of toxicity based on WASp degradation in activated PBMCs is increased cytotoxic effect of SMC13 on cells stimulated with PHA (as in figure 3B positive control but also treated with SMC13). This has not been determined or shown."

Based on our experience in studying lymphocyte activation, we believe that there is a fundamental difference in lymphocyte activation during infection vs. the constitutive activation in lymphoproliferative diseases. We postulate that SMC#13 more strongly impacts cancerous cells as compared to healthy activated cells due to the time window of its activity, i.e. the period during which WASp is open and unprotected by the WIP chaperon is significantly larger in transformed cells. Obviously, in order to advance SMC#13 toward clinical testing as a therapeutic agent for hematological malignancies, these issues should be further investigated. Moreover, based on our data, SMC#13 does not aggravate cytokine secretion.

THIS REVIEWER AGREES THAT FURTHER RESEARCH IS NEEDED TO ADDRESS THIS ASPECT, THAT DOES NOT NECESSARILY HAVE TO BE COVERED IN A SINGLE MANUSCRIPT

"The authors also show limited toxicity in healthy (naïve) PBMC to doses up to 900 uM (p13-14 I299-305 and Fig S2). However, the curve plateaus early at around 45%, has compound solubility been assessed since low solubility might confound the TD50 and would explain a curve as shown in Figure S2."

Another method to evaluate cytotoxicity using PI staining is now included in the manuscript (Figure S5).

THE TD50 DETERMINED USING THE 2 METHODS IS REASONABLY SIMILAR, WHICH IS ENCOURAGING. IT WOULD NEVERTHELESS BE USEFUL TO ASSESS COMPOUND KINETIC SOLUBILITY, A QUICK AND RELATIVELY CHEAP ASSAY TO OUTSOURCE.

"4. PK-PD correlation

The i.p. PK of SMC13 is assessed and C_{max} of 97uM achieved at 100mg/kg (I341). However, this peak concentration is fleeting, and at 1h, the concentration of SMC13 has dropped to ~15uM in plasma and 10uM in tumour, and decreasing further afterwards (Fig 5A). The degradation of WASp in cells (Jurkat) is shown after 12/24 hrs incubation at 40uM, it is not clear if this short exposure to compound in vivo can actually effectively induce degradation. What the paper is lacking is time-course of degradation of WASp with SMC13 in a panel of cell lines, WASp low/high or WASp dependent/independent including those used for the in vivo experiments."

Since we achieved effective concentration of 97.2 µM in the tumors after 30 min, which is more than double the effective 40µM concentration tested in-vitro, we believe that this concentration is more than sufficient to affect WASp cellular levels. Moreover, the concentration in tumors after 30 min is below the TD50 value. The ex-vivo data presented in Figure 6A demonstrate that WASp is degraded in the tumor cells 24h after systemic injection of SMC#13. Time-course evaluation of WASp stability within short time periods presented in Figure 2E indicates that WASp already starts to degrade after a short 2h incubation. These evidences indicate that the decrease of WASp cellular concentration in the tumor cells for such a time period might downregulate WASp-dependent cellular functions for at least 24h attenuating in-vivo tumor growth. Further testing of short-term treatment regimen might be more effective for tumor attenuation and prolong mice survival.

Our data indicate that upregulation of WASp ubiquitylation and degradation in cancer cells might eventually abolish key cellular functions, thereby attenuating in vivo tumor growth.

THIS REVIEWER AGREES THAT FIGURE 6A SUPPORTS THE REDUCTION IN WASP LEVELS WITH THE CONCENTRATION OF SMC13 ACHIEVED IN VIVO.

FIGURE 2E DOES NOT PROVIDE THE ANSWER SOUGHT SINCE THE CELLS ARE ALREADY PREINCUBATED FOR 12H WITH SMC13 PRIOR TO ADMINISTRATION OF CYCLOHEXIMIDE (655-662), WHICH DOES NOT MIMIC THE PROFILE AND DURATION OF HIGH EXPOSURE OBSERVED IN VIVO. A SHORT SMC13 INCUBATION FOLLOWED BY WASHOUT OF COMPOUND/MEDIA CHANGE AND WASP LEVELS ASSESSMENT AT DIFFERENT TIME POINTS WOULD MIMIC BETTER THE IN VIVO EXPERIMENT AND SUPPORT THE DATA.

"Can the authors comment on the free drug concentration in blood vs same in cell experiments?"

Following injection of a 100mg/kg dose, the SMC concentration in the plasma was 116±20µM which is within the effective range according to our in-vitro experiments. However, as mentioned above, it is

also below the estimated TD50 value. based on all these data, we anticipate that this dosage should be sufficient according to the PK and efficacy data, but we do not rule out the possibility that shortening the intervals between the injections might improve efficacy and survival.

PLEASE NOTE THIS DOES NOT ANSWER THE REVIEWER QUESTIONS, THIS WOULD BE ADDRESSED BY DETERMINING THE PLASMA PROTEIN BINDING OF SMC13 IN FULL PLASMA AND IN PLASMA CONCENTRATION USED IN CELL EXPERIMENTS AND COMPARING THE CONCENTRATION OF UNBOUND COMPOUND IN THE TWO SETTINGS.

"Intratumoural in vivo treatment is shown in figure S6. The compound concentration following IT administration should be measured to compare IP vs IT PK and level and timing of exposure.....

The IT experiments were conducted as a proof of concept for the ability of SMC#13 to mitigate tumor growth prior to IP dosage calibration. Obviously, since the IP injections are the preferred and more common administration route, all ex-vivo experiments were based on this system.

THIS REVIEWER AGREES THAT ALTHOUGH POTENTIALLY USEFUL THIS DATA IS NOT CRUCIAL.

"The i.p. dosing is carried out at 72h difference, in 6 doses (Figure 5B-F); the remaining concentration of compound even at 24h, and even more so at 72h, will be close to zero. Unless sustained WASp degradation is shown following short exposure or the downstream effect is long-lived, it is unclear how proposed mechanism of action matches the efficacy seen. WASp reduction is shown in samples taken from tumours treated with SMC13 (Fig S6E, Fig 6A), how long after dosing is the sample taken – this should be clarified."

Please see above, point#4, our answer to this question. In addition, the details regarding the time of tumor extraction are now mentioned on page 18, lines 415-416. Furthermore, we anticipate that an additive long term functional effect occurs after sequential exposures of the cancerous cells to SMC#13.

THE ADDITIONAL INCLUSION OF TIMELINES SINCE TREATMENT IS WELCOME AND ADDRESSES THE QUESTION.

"5. Target validation

A key set of data lacking from this paper is the orthogonal genetic validation of target with (inducible) siRNA of WASp or WIP, mutagenesis of the Lys76/81 or of residues critical for binding between WIP/WASP. Is any of the phenotypes shown in the Jurkat/Raji cells reproduced by knockdown? For example, reduced migration, increased apoptosis, Ki-67 marker, reduced cell proliferation, dependence on activation of PBMC?"

We now include orthogonal validation using the WASp Δ WH1 mutant in MST, BLI and biochemical analysis (Figures 1 and 2D). These results show that WASp lacking the WH1 domain does not bind SMC#13 and is resistant to degradation.

We currently have data regarding the effects of WASp silencing in hematological malignancies; however, we anticipate publishing these results as part of a separate manuscript. Indeed, as noted by the reviewer, knockdown of WASp using siRNAs results in similar effect as SMC#13. Indeed, Murga-Zamalloa et al, 2016 showed that WASp expression correlates with severity of anaplastic large cell lymphoma. Similar results are shown in Figure S9.

THE ADDITIONAL DATA SUPPORT THE BIOCHEMICAL MECHANISM AND IMPROVES THE MANUSCRIPT. REGARDING THE TARGET VALIDATION ASPECT, EVEN LIMITED PRELIMINARY GENETIC DATA ACQUIRED BY THE AUTHORS, WOULD BE USEFUL TO CONVINCING A READER THAT THE MECHANISM SHOWN IS THROUGH ON-TARGET RATHER THAN OFF-TARGET EFFECTS.

Point by point letter #NCOMMS-20-10509 entitled “Targeting WASp Provides a Therapeutic Approach for Hematopoietic Malignancies”.

Following the reviewer’s requests, we address the concerns with the information and data below:

I. Virtual mutagenesis was performed at the following locations. For clarity, positions are given with reference to human WASp, and the base residues used are those of 2IFS.

At position 71, Valine was replaced with Alanine.

At position 72, Alanine was replaced with Valine.

At position 74, Leucine was replaced with Phenylalanine.

At position 81, Arginine was replaced with Lysine.

This information was added to the Materials and Methods section of the manuscript (lines 576-577).

II. We agree with the reviewer; describing the full calculations that led us to select SMC#13 requires a separate publication. Most importantly, the focus of this manuscript is not the description of new docking scoring functions, but rather the development of a novel therapeutic approach – based on the enhancement of the natural degradation of activated-WASp, resulting in the abolishment of WASp-dependent activities, subsequently leading to cancer cell death. This is evident in the multiple and parallel methodologies utilized in both in-vitro and in-vivo systems. We provide all the details required to reproduce the relevant results: the biochemical and cell-based assays, the in-vivo studies, and the full molecular and commercial description of the proteins and reagents used, including SMC#13 (the full chemical structure, and the manufacturer’s statement confirming the source, identity, and purity of the compound, are disclosed in line 570). Importantly, SMC#13 is commercially available. We show that SMC#13 binds the target and has the desired effect in-vitro and in-vivo. We avoid making claims regarding the power, generality or performance of the algorithms used for discovering SMC#13. Nevertheless, we provide further details (please see pages 27-28) of the in-silico process, describing how we identified SMC#13 within the length constraints. Given that, as correctly noted by the reviewer, the detailed computational methodology requires an independent manuscript, its brief and reproducible description was nevertheless added.

III. It should be noted that the model presented was generated strictly for the purposes of illustration and that the scoring process (and the virtual screening of leading candidates) was not dependent on it.

We thank the reviewer for his/her comment, and have updated the putative model. The current model exhibits the same key characteristics of the previous one, namely the existence of multiple non-covalent bonds with WASp, its spatial overlap with WIP, and avoiding blocking the region suggested to be necessary for the WASp ubiquitylation. The text was modified to further highlight the fact that the support the model lends to our mechanistic hypothesis is putative (line 476; line 482).

IV. We provide a certificate of SMC#13 analysis from ChemBridge including H-NMR, LC-MS spectra, and purity of the SMC which is >90% according to the manufacture company (attached at the end of this letter).

V. As requested by the reviewer, we enclose data regarding experimental measurement of SMC#13 solubility (please refer to the data figure “solubility” attached at the end of this letter). using the well-established method based on octanol-water distribution coefficient. For the aqueous phase we used PBS with a physiological pH of 7.4 and measured the log D (distribution coefficient) value as ~3.3 via HPLC, which verified the predicted value of ~3.8 using SwissADME (presented in Table 1).

VI. The pharmacokinetics of SMC #13 indicate that the tumor cells are exposed to the molecule for roughly 1 hour after injection, with maximal concentration occurring 30 minutes post injection (with a 90µM peak concentration).

As requested by the reviewer, we conducted a short in vitro SMC#13 incubation for 30 min followed by washout of the media, and measured WASp levels at different time points. As shown in new Fig.S3D, short incubation and washout of media resulted in a reduction of WASp intracellular concentration. These results corroborate the *in-vivo* data. As seen in Figure S3D, WASp undergoes degradation after exposure to SMC #13, with significant degradation becoming measurable 8 hours after treatment.

From the ex-vivo experiments, we see that tumor cells 24 h after injection exhibit inhibited WASp dependent-activity, as indicated by substantial decrease of WASp expression, cellular activation, proliferation, migration, and invasion (Fig. 6A-E). From these results we conclude that WASp expression is likely downregulated for a time window of between 8 and 24 h at the very least, and WASp dependent-activity is disrupted for the duration of this time window during the 72 h injection intervals.

VII. We performed a plasma protein binding assay of SMC #13, evaluating its binding to full plasma proteins and in-vitro cultured media supplemented with serum (please refer to the data figure “plasma protein binding” attached at the end of this letter). Following 30 minute incubation of SMC #13 with full plasma, medium, or PBS as a control, unbound SMC was separated from the protein bound SMC #13 complex using a molecular weight cutoff of 10kD and measured using HPLC as detailed in the Materials and Methods. One technical modification was the change in flow rate from 0.5ml/min to 0.35ml/min which

shifted the retention time to 13 min. The calculated plasma protein binding of SMC #13 is approximately 85.9%, while in-vitro media supplemented with 10% FCS yielded 78% binding. Furthermore, we also evaluated the plasma protein binding of SMC#13 to a value of 85% using "preADMET", a virtual predictor (Lee et al. The PreADME Approach: Web-based program for rapid prediction of physico-chemical, drug absorption and drug-like properties, <https://preadmet.bmdrc.kr/>).

VIII. As requested by the reviewer, we added evidence indicating that the mechanism shown is a true on-target effect (new Fig. S3E). In this figure we show that CRISPR/CAS9-mediated knockout of endogenous WASp in cells results in a heavily reduced cell displacement (from 23.9 to 5.2 μm ; a 4.6-fold decrease). A similar trend was observed in ex-vivo experiments, by extracting blood cancer cells from mice treated with SMC #13 (Fig. 6D); a 4-fold decrease was seen in their migration (in comparison to the control mice treated with vehicle).

This observation is similar to the well described phenotype of WASp deficient cells and corroborates the results of SMC#13 treatment shown in the manuscript.

We hope that based on these revisions, our manuscript will now be found suitable for publication in Nature Communications.

Solubility

A

B

Plasma Protein Binding

A

B

C

PT01017601

ID	94281174	461.6095	C ₂₈ H ₃₅ N ₃ O ₃
----	----------	----------	---

Data File D:\DATA\MICRA\2AA-1401.D
 Sample Name: PT010176P2-A-01
 Instrument 1 15/04/2020 11:49:58 4
 Column: ONYX MONOLITHIC, C18 50x4.6mm | 3.75ml/min | Columns Reg Valve
 Gradient: "A"->@2.0min->"B"(Hold 0.6min)->@0.2min->"A"->PostRun
 PMP1, Solvent A : 0.1%TFA, 2.5%AcN in H2O
 PMP1, Solvent B : 0.1%TFA in AcN
 PMP1, Solvent C : --NOT USED--
 PMP1, Solvent D : --NOT USED--
 Ionization mode : API-ES Positive

Signal 1: ADC1 B, ELSD

Peak #	RetTime [min]	Type	Width [min]	Area [mV*s]	Height [mV]	Area %
1	1.597	MM	0.0410	252.21652	102.43323	100.0000
Totals :				252.21652	102.43323	

Signal 2: DAD1 A, Sig=300,200 Ref=off

Peak #	RetTime [min]	Type	Width [min]	Area [mAU*s]	Height [mAU]	Area %
1	1.547	MM	0.0468	948.99445	337.95166	100.0000
Totals :				948.99445	337.95166	

Signal 3: MSD1 TIC, MS File

Peak #	RetTime [min]	Type	Width [min]	Area	Height	Area %
1	1.617	MM	0.0709	1.24513e7	2.92898e6	100.0000
Totals :				1.24513e7	2.92898e6	

ID	94281174	461.6095	C ₂₈ H ₃₅ N ₃ O ₃
----	----------	----------	---

REVIEWER COMMENTS

Reviewer #4 (Remarks to the Author):

Please see my comments to each of the authors' answers in the "Point by point letter #NCOMMS-20-10509 entitled "Targeting WASp Provides a Therapeutic Approach for Hematopoietic Malignancies"."

Point I (residues for virtual mutagenesis)

This has been addressed as requested, note it is shown on lines 588-589 not lines 576-577.

Point II (description of computational methods)

Addition of further information regarding the computational aspect is welcome.

Point III (modelling in Fig. S1B-S1C)

This reviewer agrees that the updated model corrects the deficiencies of the previous model around the indane system, however other angles, in particular the 4-methoxy group in Fig S1C and also to some extent in Fig S1B, are now out of plane with the aryl ring of the 4-methoxybenzyl moiety, unlike the H atoms and the methylene group which are correctly in the same plane. I suggest that the authors either correct this or remove Figures S1B-C, which as they point out are not influencing the scoring process.

Point IV (Analytical data for compound SMC#13)

The NMR and LCMS data are in agreement with the structure, and the purity looks reasonable based on analytical data provided.

Point V (physicochemical data)

The measured logD is an useful benchmark for the predictive software used. The authors should report the measured solubility and log D values where these are required to support the data (e.g. on lines 503-504) rather than using qualitative sentences such as "might be explained by moderate solubility of the SMC and high lipophilicity", since this was the purpose of assessing them.

Point VI (PK-PD correlation)

This data supports the hypothesis, even if there is about 50% WASp left at 8 hrs, it is possible that in vivo there is compound left for longer that would provide a stronger degradation. The caption for Fig S3D should mention that the incubation was done for 30 min, as per the answer in the letter.

Point VII (Plasma protein binding)

This is very useful data, and the respective unbound fractions of 14.1% vs 22% in plasma vs culture media suggest a drop-off in unbound compound of only about 30% between cell data and in vivo data, which is consistent with the effect observed. This data should be reported in the manuscript.

Point VIII (Genetic validation)

This data support the authors' hypothesis and strengthen the argument. The authors should add the caption for Figure S3E, which is currently missing.

Overall conclusion

Most of the points raised by this reviewer have been addressed. If the authors address the remaining

points (correction of model in Figure S1B, S1C as discussed above or alternatively removal of figures; addition of time of incubation in caption S3D; providing the solubility and logD values in text; reporting the plasma protein binding; adding the caption for Figure S3E), the manuscript is suitable for publication.

Point by point letter #NCOMMS-20-10509 entitled “Targeting WASp Provides a Therapeutic Approach for Hematopoietic Malignancies”.

We thank the reviewer for his/her constructive comments. We addressed all the comments with no exception, as detailed below:

- Point I (residues for virtual mutagenesis)-the text was corrected.
- Point II (description of computational methods)-Additional information regarding the computational process was added to the manuscript (Lines 159-170).
- Point III (modelling in Fig. S1B-S1C)-The figure S1B-C was removed as suggested by the reviewer.
- Point IV (Analytical data for compound SMC#13)
-We thank the reviewer for his/her positive comment.
- Point V (physicochemical data) -The logD value was incorporated in the text (lines 512-514 and Fig. S1D) in accordance with the reviewer's comment.
- Point VI (PK-PD correlation)- The caption was changed according to the reviewer's comment and the required details were added.
- Point VII (Plasma protein binding)-We appreciate the positive input of the reviewer regarding the data on the plasma protein binding. This figure is now incorporated in the manuscript (lines 420-428 and new Fig. S9).
- Point VIII (Genetic validation)-The missing caption was added to the text lines 1374-1379.

All the additions and the corrections were made as suggested by the reviewer and we hope that based on these revisions, our manuscript will now be found suitable for publication in Nature Communications.